# Reconstitution of contractile actomyosin rings in vesicles

Thomas Litschel [1], Charlotte F. Kelley [1,2], Danielle Holz[3], Maral Adeli Koudehi[3], Sven K. Vogel[1], Laura Burbaum[1], Naoko Mizuno [2], Dimitrios Vavylonis[3] & Petra Schwille [1✉]

One of the grand challenges of bottom-up synthetic biology is the development of minimal machineries for cell division. The mechanical transformation of large-scale compartments, such as Giant Unilamellar Vesicles (GUVs), requires the geometry-specific coordination of active elements, several orders of magnitude larger than the molecular scale. Of all cytoskeletal structures, large-scale actomyosin rings appear to be the most promising cellular elements to accomplish this task. Here, we have adopted advanced encapsulation methods to study bundled actin filaments in GUVs and compare our results with theoretical modeling. By changing few key parameters, actin polymerization can be differentiated to resemble various types of networks in living cells. Importantly, we find membrane binding to be crucial for the robust condensation into a single actin ring in spherical vesicles, as predicted by theoretical considerations. Upon force generation by ATP-driven myosin motors, these ring-like actin structures contract and locally constrict the vesicle, forming furrow-like deformations. On the other hand, cortex-like actin networks are shown to induce and stabilize deformations from spherical shapes.

[1] Department of Cellular and Molecular Biophysics, Max Planck Institute of Biochemistry, Martinsried, Germany. [2] Department of Structural Cell Biology, Max Planck Institute of Biochemistry, Martinsried, Germany. [3] Department of Physics, Lehigh University, Bethlehem, PA, USA. ✉email: schwille@biochem.mpg.de

In cells, actin filaments are organized into cross-linked, branched, and bundled networks. These different architectures appear in structures, such as filopodia, stress fibers, the cell cortex, and contractile actomyosin rings; each has unique physical properties and fulfills different roles in important cellular processes[1]. These different structures must be actively assembled and maintained by cellular factors, such as the many actin cross-linking proteins. By mediating higher-order actin organization, cross-linkers allow actin filaments to fill a diverse array of structural and functional roles within cells[2,3].

In many cases, actin networks are linked to, or organized around, cellular membranes. Actin polymerization is a driving force behind many examples of membrane dynamics, including cell motility, membrane trafficking, and cell division[1]. Many of the actin-binding proteins involved in these processes are directly regulated via interactions with phospholipid bilayers[4,5] and membrane interactions have, in turn, been shown to physically guide actin assembly[6]. While the link between the actin cytoskeleton and phospholipid bilayers is clear, how these connections affect the large-scale organization of complex actin networks remains an open question.

Actin is not only one of the most prevalent proteins in current reconstitution experiments[7,8], but was also one of the first proteins to be explored in such approaches[9,10]. The focus of actin-related work has since shifted from identifying the components responsible for muscle contraction[11], to investigating more detailed aspects of the cytoskeleton[8,12], such as the dynamics of actin assembly[13,14] or the cross-talk with other cytoskeletal elements[15]. These experiments have extended to actin–membrane interactions, including reconstitution of actin cortices on the outside of giant unilamellar vesicles (GUVs)[16–18], and contractile actomyosin networks associated with supported membranes[19–21]. Recently, creating a synthetic cell with minimal components recapitulating crucial life processes, such as self-organization, homeostasis, and replication, has become an attractive goal[22,23]. As such, there is increased interest in work with actin in confinement and specifically within GUVs[24,25], in order to mimic cellular mechanics, by encapsulating actin and actin-binding proteins in vesicles[26–29]. However, the investigation of higher-order actin structures or networks has been the subject of few studies thus far[29–32].

Particularly, interesting for the reconstitution of actin-related cell processes is the co-encapsulation of myosin with actin, in order to form contractile actomyosin structures. While Tsai et al. showed the reconstitution of a contractile network in vesicles[26], and others have reconstituted actomyosin networks in vesicles that imitate actin cortices[27,29], contractile actomyosin rings have proven difficult to achieve. A true milestone toward the reconstitution of a division ring is the work of Miyazaki and coworkers, who encapsulated actomyosin with a depletant in water-in-oil droplets[33]. They showed not only that the formation of equatorial rings from actin bundles is a spontaneous process that occurs in spherical confinement, in order to minimize the elastic energy of the bundles, but also demonstrate the controlled contraction of these actomyosin rings.

Due to the difficulty of encapsulating functional proteins within membrane vesicles, much of the past work has been limited to water-in-oil emulsions and adding proteins to the outside of vesicles or onto supported lipid membrane systems. However, novel encapsulation methods such as continuous droplet interface crossing encapsulation (cDICE), as used here, have enabled the efficient transfer of proteins and other biomolecules into cell-sized phospholipid vesicles, as an ideal setting to study complex cellular processes involving membranes[34–37]. The challenges and applications of protein encapsulation in GUVs are summarized in a current review article[38]. Here, we optimized actin encapsulation for a high degree of reproducibility and precision. This allowed us to reconstitute novel cell-like cytoskeletal features, as well as compare our experimental results with numerical simulations of confined interacting actin filaments. The development of experimentally testable predictive theoretical models is central for the future design of complex experiments that approach the functional complexity of biological systems.

We combined actin bundling and actin–membrane linkage to obtain results more closely resembling in vivo morphologies than previously achieved in vitro. Specifically, we induced the formation of membrane-bound single actin rings, which imitate the contractile division rings observed in many cells. In agreement with our numerical simulations, we show that membrane anchoring significantly promotes the formation of actin rings inside vesicles. We achieved close to 100% probability of ring formation in vesicles when using the focal adhesion proteins talin and vinculin, which we recently identified as effective actin bundlers[39]. With the inclusion of motor proteins, these actomyosin rings contract similar to those observed in yeast protoplasts[40].

Thus, in this study, we not only achieve the formation of membrane-attached actin rings within lipid vesicles, but also observe large-scale membrane deformation when including myosin in the system. Although aspects of our study were previously addressed individually, such as encapsulation of actin bundles, actin binding to the inner membrane leaflet of a vesicle, or encapsulation of contractile actomyosin networks in vesicles, until now it proved too experimentally challenging to reproducibly combine these within one experimental system. Our results provide a high-yield approach, returning reproducible and quantifiable results, that brings us that much closer to the ultimate goal of being able to quantitatively design and experimentally achieve full division of a synthetic membrane compartment, and thus, to the self-reproduction of artificial cells, a persistent goal in bottom-up biology[41–44].

## Results

**Experimental system.** In order to investigate the interplay between actin cross-linking and membrane binding, we used a modified cDICE method[45,46] to encapsulate G-actin with associated proteins and generate cytoskeletal GUVs made from the lipid POPC (Fig. 1a). Since components cannot be added once the reaction mix is encapsulated, the precise composition of the initial reaction mix is crucial. By tuning concentrations of the polymerization buffer, bundling proteins, membrane anchors, and motor proteins, we manipulated the final morphology of the actin network.

By co-encapsulating actin with known actin cross-linking proteins, we achieved large-scale networks with clearly discernible actin structures, similar to earlier studies with cytoskeletal GUVs[30]. We tested four different types of actin bundling proteins: fascin, α-actinin, vasodilator-stimulated phosphoprotein (VASP), and a combination of the focal adhesion proteins talin and vinculin. Each case represents a slightly different mechanism of actin binding. Fascin, a 55 kDa protein, binds to actin through two distinct actin-binding sites, thereby inducing filament cross-links as a monomer[47]. α-Actinin (110 kDa) forms a dimer which bridges two filaments[30,48]. Talin (272 kDa) and vinculin (116 kDa) both dimerize, and also require interactions with each other in order to bind and bundle actin filaments[39]. Here we use a deregulated vinculin mutant (see Supplementary Information). VASP (50 kDa) forms a tetramer, which can link up to four filaments together[49]. Under all four conditions, the formation of thick filament bundles was observed (Fig. 1b, c). Interestingly, while α-actinin, talin/vinculin, and VASP all produced similar

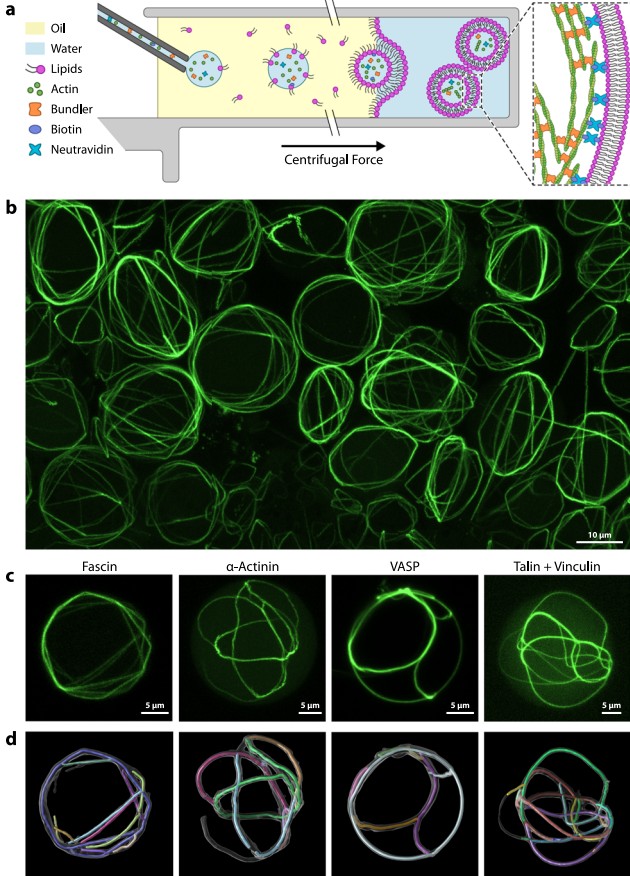

**Fig. 1 Encapsulation of bundled actin in giant unilamellar vesicles. a** Schematic depiction of the vesicle generation process. The aqueous protein solution is injected into a rotating chamber through a glass capillary. Droplets form at the capillary tip in the oil phase, which contains lipids. The droplets then pass through a water–oil interphase lined with a second lipid monolayer, forming the giant unilamellar vesicles (GUVs). **b** Field of view image (Z-projections of confocal stacks) with many cytoskeletal vesicles. Actin in green. See Supplementary Movie 1 for 3D effect. **c** Comparison of cytoskeletal vesicles with actin bundled by four different types of bundling proteins. We used 2 μM actin in all cases, but due to differences in bundling activity, different concentrations of bundling protein: 0.3 μM fascin, 0.9 μM VASP, 1 μM α-actinin, 2 μM talin, and 2 μM vinculin. **d** Automated tracing of bundles by analysis script. Confocal z-stacks are converted into a three-dimensional "skeleton" model. Supplementary Movie 2 shows a 3D view of both representations of these vesicles.

morphologies, fascin bundles take on the most unique appearance. These bundles often bend in kinks when their path is obstructed by the membrane, while the other proteins form smoothly curved bundles that can follow the curvature of the encapsulating membrane (Fig. 1c, d). While similar observation have been made in the past, generally, bulk assays will be better suited for quantifying these bundle parameters[30,50–53].

After establishing successful encapsulation of actin and its bundling proteins, we modified the approach and linked the actin filaments to the phospholipid bilayer via biotin–neutravidin bonds, similar to previous work on planar supported lipid bilayers[19]. This requires the incorporation of biotinylated lipids in the vesicle membranes and the addition of both biotinylated g-actin and neutravidin in the encapsulated reaction mix. We tested different fractions of biotinylated lipids, as well as biotinylated actin (Supplementary Figs. 2 and 3) and identified 1%

biotinylated lipids and 4% biotinylated actin as suitable amounts, which we used in the following experiments.

**Numerical simulations**. Theoretical predictions by Adeli Koudehi et al. have suggested that actin organization depends crucially on confinement and surface attachment[54]. In order to explore the agreement of our experimental results with these simulations, we adopted their theoretical model. As such, we performed numerical simulations of interacting actin filaments under spherical confinement using Brownian dynamics (see Supplementary Methods)[54]. Semi-flexible actin filaments were modeled as beads connected by springs, with cross-linking represented by a short-range attraction with spring constant $k_{atr}$. Polymerization from an initial number of filament seeds was simulated by addition of beads at one of the filament ends (representing the barbed end). The number of seeds was changed to achieve different final filament lengths. Boundary attraction was simulated as short-range attraction to the confining boundary. Simulated maximum intensity projections were performed as in Bidone et al.[55].

**Membrane attachment shapes actin organization by curvature induction**. We performed a series of experiments with the simplest bundling protein fascin to investigate the effects of membrane binding on bundle morphology. We notice that membrane-binding primarily affects the curvature of these bundles: while actin with fascin forms very straight bundles that are just generally confined by the membrane, we see that membrane-bound fascin bundles often adopt the exact curvature of the membrane (Fig. 2a). Figure 2b shows a histogram of the distribution of bundle curvatures in these vesicles. The histogram shows a much broader distribution for unbound bundles, with a maximum at low curvatures, while the maximum for membrane-bound actin bundles is centered around the curvature of the membrane (relative curvature = 1.0).

Despite this difference on a small scale, the general distribution of bundles within the vesicles seems to be largely independent of the presence of actin–membrane linkers. Figure 3a shows a set of conditions with and without membrane binding. We quantified the average actin distribution for each condition and find that (with some exceptions) actin is consistently positioned in close proximity to the membrane, with only minor differences between conditions with and without membrane linkers (Fig. 3c, d). While in previous work even unbundled actin has been seen to be more concentrated at the membrane[32], our results indicate that bundles need to be sufficiently long, so that confinement by the vesicle boundary forces them to bend and concentrate at the inner surface. We note that for 2 μM actin (Fig. 3a, top row) even low concentrations of fascin are sufficient to cause this effect, while we do observe that the thickness of the bundles increases with higher concentrations (Supplementary Fig. 5).

At higher actin concentrations (6 μM) and low fascin to actin ratios (3.3 and 6.7%), bundles were shorter and thus more homogeneously distributed in the vesicles when not bound to the membrane (Fig. 3d). Interestingly, we note that membrane-binding affects the threshold at which long actin bundles form: at a fascin to actin ratio of 6.7%, we only observe long bundles when we include membrane linkers (see also Supplementary Fig. 6). These observations agree with corresponding simulations (Fig. 3b).

In our experiments, ring-like structures consistently form at 2 μM actin, while at 6 μM actin, multiple bundles usually arrange themselves into cortex-like structures that do not condense into single rings. In our simulations, we see similar cortex-like morphologies in the early stages, but at longer times these

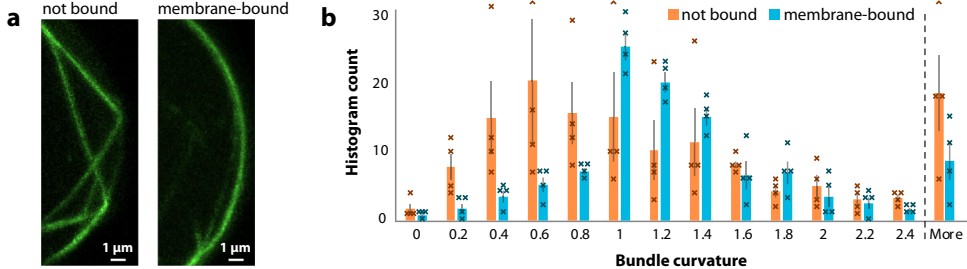

**Fig. 2 Membrane attachment affects curvature of actin bundles. a** Each image shows a section of the membrane of a GUV with actin bundles cross-linked by fascin (2 μM actin and 0.2 μM fascin). Actin is in green. Unattached bundles have long sections with low curvature, while membrane-bound bundles follow the curvature of the membrane. **b** Distribution of curvatures for actin bundles in vesicles with and without membrane binding. $n = 4$ GUVs for each condition. Confocal $z$-stacks were converted into 3D information (see Fig. 1d). Bundles were then divided into small segments and their curvature was measured. Curvature is normalized to membrane curvature, so that a curvature of 1.0 equals the curvature of the membrane. Data are shown as mean values with individual histogram counts and SEM. More details about the analysis can be found in the Supplementary Method.

condense into rings, both for low and high actin concentrations (Supplementary Fig. 7). As discussed below, the smaller confinement size we chose due to computation limitations favors ring formation. Another ring promoting factor could be the absence of a simulated maximum bundle thickness, unlike in experiments[56].

**Ring formation**. Excitingly, and in accordance with theoretical predictions, the most noticeable effect of membrane attachment was an increase in the formation of single actin rings. Although ring formation could still be observed without membrane–actin links, the introduction of membrane binding greatly enhances the probability of actin condensation into one single clearly discernible ring in vesicles. Membrane-bound actin rings have so far not been reported within synthetic vesicles.

Figure 4 highlights this effect of membrane attachment on the formation of actin rings. Figure 4a summarizes ring formation probabilities for three different bundlers, comparing conditions with and without membrane binding. We chose actin and bundler concentrations for which the formation of single rings is already relatively likely (12–35%) even without membrane attachment. In vesicles with membrane-attached actin, probability of ring formation roughly doubles for all bundlers, and reaches up to 80% for actin bundled by vinculin and talin (fluorescence image in Fig. 4c).

In simulations, Adeli Koudehi et al. found that boundary attraction in the case of spherical confinement enhances the probability of ring formation from bundled filaments[54]. However, the effect of boundary attraction in their work was studied for filament lengths larger than the confining diameter, whereas in experiments, we observed increased ring formation for vesicles and actin concentrations where the opposite should be true. We thus performed new simulations of actin filaments for concentrations chosen as in our experiments ($c = 2$ μM), and varied their lengths and confinement sizes (Fig. 4b, Supplementary Fig. 11, and Supplementary Movie 5). We selected filament cross-linking simulation parameters that lead to bundle formation without filament sliding and associated bundle compaction $k_{atr} = 2$ pN/μm. Including surface attraction greatly enhanced ring and ring-like structure formation for short filaments (length $L = 1.2$ μm) in small spheres (radius $R = 2.5$ μm). We also observed an enhancement of ring formation for filament lengths and sphere sizes comparable to that of our experiments ($L = 6$ μm as estimated from prior studies[33,57,58], $R = 5$ μm), including when we increased the persistence length of individual actin filaments to simulate cross-linking induced bundle stiffening (Supplementary Fig. 11). Inspired by modeling results implying that the probability of ring formation depends on compartment size[54], we analyzed our

experimental data to confirm that rings preferably form in smaller vesicles (Supplementary Fig. 12).

**Actomyosin contraction**. The rings observed here can be assumed to mimic reorganization of actin that occurs during the last stages of cell division. In order to take this analogy one step further, we included muscle myosin II with the ultimate goal of forming a contractile actomyosin ring. Constriction of non-anchored actin rings was shown by Miyazaki et al., who demonstrated myosin-mediated contraction in a less cell-like system. They used actin bundled by depletion forces in water-in-oil droplets and showed that the behavior reproduced by this system has a striking resemblance to constricting cell division rings[33].

In our vesicles, the addition of myosin complicated the formation of single actin bundle rings. We used low concentrations of myosin II, such that the effect of myosin activity on bundling was minimized and motor-induced constriction slow enough to be observed while imaging. Although it appeared that the appropriate assay conditions for homogeneously contracting single rings have not yet been met in our giant vesicles, we recorded the constriction of a membrane-anchored ring-like structure along with membrane deformations. Figure 5a and Supplementary Movie 6 show this example over the course of 2 h. In accordance with our expectations, in this minimal system without further ring-stabilizing components, the constricting actomyosin ring eventually slides along the membrane and collapses into a single condensate on one side of the vesicle, a behavior that has been seen in yeast cells lacking cell walls[40]. Such an arbitrary local collapse is not too surprising, as coordinated ring constriction in the cell is a highly spatially regulated process involving hundreds of proteins. Clearly, additional cellular machinery is required to stabilize the position of the ring, and membrane geometry and fluidity likely play additional roles. Figure 5a shows how the actomyosin ring initially deforms the vesicle membrane (orange arrows), leading to a furrow-like indentation. The entire time series without overlays is shown in Supplementary Movie 6. Our experiments clearly show that the actin bundles are firmly attached to the inner leaflet of the vesicle membrane and that active forces are exerted by the motor proteins, capable of deforming the vesicle. Figure 5b shows another instance of actomyosin contractions leading to deformations of the vesicle membrane. Unfortunately the above mentioned complications in assay design hampered consistent observations of these membrane deformations.

An additional effect of the contraction of membrane-bound actomyosin is a change in the $x$–$y$ cross section area of the vesicle after contraction. This effect also appears for vesicles without

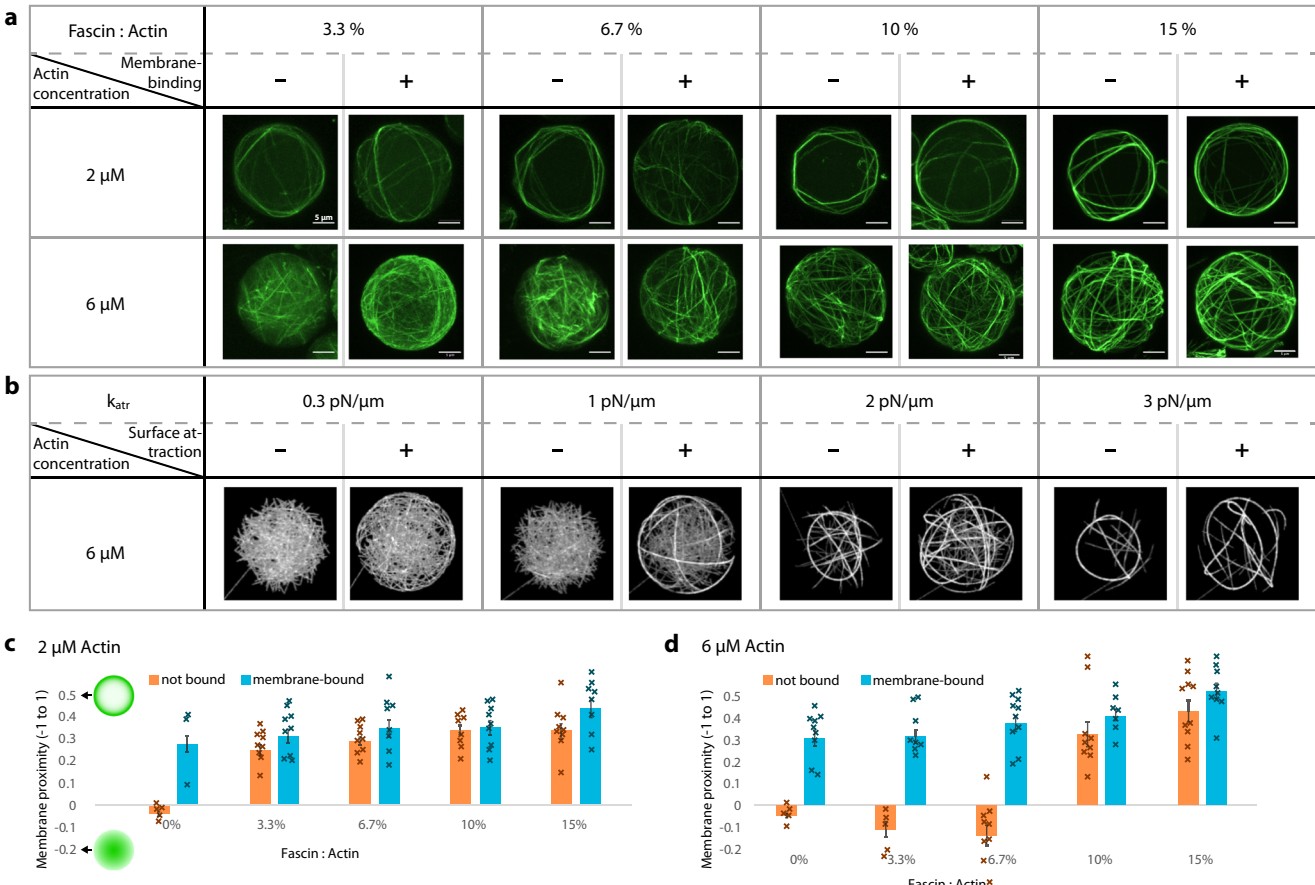

**Fig. 3 Actin organization in dependence on bundler to actin ratio and membrane binding.** Actin is positioned close to the membrane regardless of membrane-anchoring. **a** Overview of conditions with varying actin concentration (2 µM and 6 µM), fascin to actin ratios (3.3:100–15:100) and with and without actin–membrane binding. Supplementary Movie 3 shows a 3D version of this figure panel. **b** Simulations for 6 µM conditions as in **a** using radius = 6 µm. **c** Average membrane proximity of actin signal for vesicles with 2 µM actin. Normalized range from −1 (all actin in the center of the vesicle) to +1 (all actin on the membrane). **d** Membrane proximity of actin signal for vesicles with 6 µM actin. Data in **c** and **d** are presented as mean values with individual data points and SEM. $n = 10$ vesicles for each condition (except $n \geq 3$ for controls with 0% fascin).

initial furrow constriction. This is likely due to crumpling of the membrane into the actomyosin contraction point, which decreases membrane area (while vesicle volume is largely preserved) and increases membrane tension. As a result, vesicles that are initially slightly deflated become more spherical as a result of actomyosin contraction. Supplementary Figure 14a shows a differential interference contrast (DIC) image of the actomyosin contraction point in Fig. 5a, and more examples of vesicles with shrinking $x$–$y$ cross sections.

**Shaping the membrane compartment.** Lipid membranes are highly flexible, and the shape of GUVs is mostly determined by the osmotic pressure inside the vesicle with respect to its environment. If this pressure is low, i.e., vesicles are osmotically deflated, strong deviations from the spherical shape are possible, and additional mechanical determinants, such as external forces or an encapsulated cytoskeleton induce arbitrary shapes of the vesicles[32], as can be seen in Fig. 6a. The experiment we performed in Fig. 6b confirms the role of an artificial cytoskeleton in determining vesicle shape, i.e., stabilizing the shapes of membrane compartments: by imaging deformed cytoskeletal vesicles with increased laser power on our confocal microscope, the actin filaments depolymerize after some time due to photodamage[59], relaxing the cytoskeleton-inferred shape determinants and leaving the deflated vesicles without internal support. This leads to

dramatic changes in their shape, usually by taking on a spheroid (oblate) shape.

These stabilizing cortices of actin bundles can even protect the vesicles, e.g., against the unfavorable conditions of sample preparation for cryo-electron microscopy, specifically the drying of the sample (removal of the surrounding aqueous phase): Fig. 6c shows a cryo-scanning electron microscopy image of frozen cytoskeletal vesicles. When trying to freeze and image vesicles without an encapsulated actin cortex or with actin bundles that are not attached to the membrane, vesicles rarely survive the process (Supplementary Fig. 15). Although a thorough quantitative assessment of this effect is not within the scope of this study, it confirms previous work that GUVs can be stabilized through a shell of cross-linked material on the membrane of GUVs, not only with unbundled actin[60], but also with other proteins[61], as well as DNA origami[62]. Here, we show that heterogeneously distributed, higher-order structures can potentially achieve a similar mechanical effect.

## Discussion

In this work, we succeeded in reconstituting ring-like actomyosin structures in GUVs. With respect to a suitable protein machinery that may serve as a minimal divisome for protocells, this constitutes the first important step toward assembling contractile rings of sufficiently large sizes. To this end, we encapsulated a reaction mix into the vesicles that causes actin to polymerize,

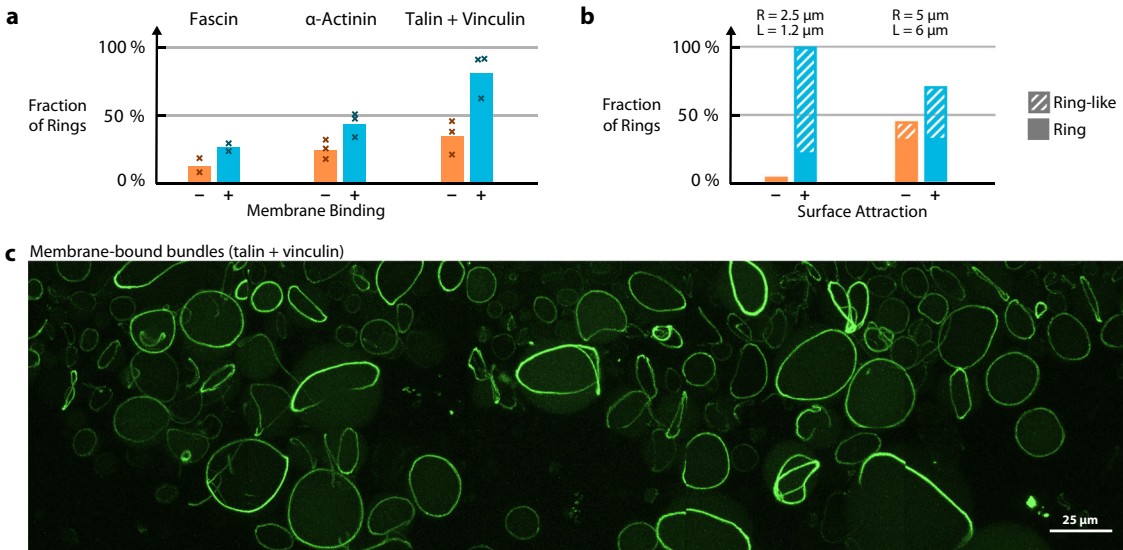

**Fig. 4 Formation of membrane-anchored actin rings. a** Membrane-binding promotes ring formation. Shown is the probability of the formation of single actin rings in GUVs (i.e., GUVs with one single unbranched actin bundle connected into a ring) in the absence or presence of membrane-anchoring. Data are shown as mean values with individual fractions of experimental runs. Three hundred ninety two vesicles between 15 and 20 μm were analyzed in $n = 2$ (fascin) or $n = 3$ experimental runs per condition. We use 2 μM actin in all cases, but due to differences in bundling activity different concentrations of bundling protein: 0.3 μM fascin, 1 μM α-actinin, 2 μM talin, and 2 μM vinculin. **b** Probability of ring formation for simulations with different initial parameters ($R$: vesicle radius, $L$: filament lengths). In simulations, we classified rings with small gaps or closed rings with additional side branches as "ring like" (see Supplementary Method). Snapshots from all simulation shown in Supplementary Fig. 11. **c** Condition with particularly robust ring formation: actin bundled by talin with vinculin and bound to the membrane. Supplementary Movie 4 shows a 3D view of this image. Supplementary Figure 8 shows a DIC image of this field of view, and Supplementary Figs. 9 and 10 show rings formed by other bundling proteins.

bundle, bind to the vesicle membrane, and even contract. We show that the bundle networks can be highly organized and, under many conditions, reproducibly cross-linked into single rings.

Clearly discernible ring formation has been previously shown by Miyazaki et al., who utilized depletion forces through the crowding agent methylcellulose, while confining actin polymers in small water-in-oil droplets[33]. These rings were shown to contract, but due to the lack of surface attachment in this system, unable to transmit contractile forces to the compartment interface. The formation of rings from biopolymer bundles in confinement due to a minimization of bending energy is known from both, theory[54,63] and other experimental systems[31,64,65]. Here, we achieved similar actin rings bundled by various physiological factors. However, encapsulation in lipid vesicles allowed us to go an important step further and explore the effect of attaching these rings to the compartmentalizing lipid bilayer. In this scenario, ring contraction may be able to transmit a contractile force directly to the membrane, resulting in dramatic shape transformation of the respective compartment, induced by energy dissipation within.

We have thus shown, as a proof of principle, that non-equilibrium myosin-mediated constriction of such ring-like membrane-bound actin structures can be induced in lipid membrane vesicles. These vesicle deformations (Fig. 5a) demonstrate the strength of the membrane anchoring. The final contracted state reveals myosin-induced symmetry breaking, as observed in other actomyosin in vitro systems under confinement[18,26,27,66]. For example, Tsai et al. encapsulated a contractile actomyosin system in vesicles that condensed into dense clusters[26].

Unless membrane area can be expanded at the same time, the osmotic pressure inside a spherical vesicle complicates a cell division-like symmetric constriction in the center of the vesicle, and in the absence of other geometric regulators, the fluidity of the membrane causes the ring-like bundles to "slip" and contract into a cluster in one location. Our experiments indicate, that in

order to achieve a binary fission through contraction of a single ring, more spatial determinants are required.

A behavior similar to what we observe can be seen in vivo for the contraction of actomyosin rings in yeast protoplasts (yeast cells that have been stripped of their cell walls)[40]. Stachowiak et al. beautifully demonstrated that in these spherical cells without cell walls, the contractile actomyosin ring slides along the cell membrane, collapsing into one point at the side of the cells. The absence of a cell wall in fission yeast results in both a loss of their elongated shape and a lack of stabilizing the actomyosin ring in the cell center[40].

We conclude that further assay improvement and, very likely, additional functional components will be necessary to accomplish a complete division of a cell-sized vesicle compartment in vitro. Functional studies will allow us to identify a machinery that ensures the placement of a contractile actomyosin ring at a defined site, while invoking other spatial cues to prevent the deflection of the induced contractile forces by surface slipping. The MinDE system, which has previously been shown to target FtsZ rings to the middle of compartments[67], and extend this potential of positioning even to functionally unrelated membrane-binding particles[68], may be an attractive candidate. Moreover, ring constriction could be more successful in a non-spherical, elongated vesicle; such a confinement shape, however, will likely prevent initial ring formation along the desired constriction site. Further requirements may include mechanisms to generate actin filament bundles of mixed polarity, and to sustain such distribution throughout the constriction process, possibly through turnover of components[69].

To summarize, by reconstituting a contractile actomyosin ring-like structure in GUVs, we have made one essential step forward with regard to establishing a minimal system for active membrane vesicle division from the bottom up. Using this protein machinery from eukaryotes, large-size contractile ring structures could be

Myosin II-induced contraction of membrane-bound bundles (talin + vinculin)

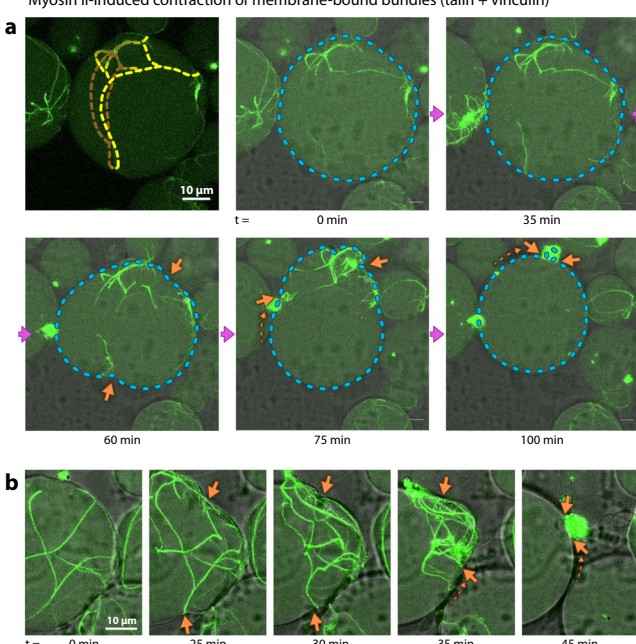

**Fig. 5 Actomyosin contraction leads to membrane deformations. a** Time series of a contracting ring-like structure in a GUV. Same conditions as in Fig. 4c but with 0.1 μM myosin II. Only the midsection of the vesicle is shown, top and bottom are missing, but the yellow dotted lines in the first frame show the approximate position of the bundles (see also Supplementary Fig. 13). Cyan dotted lines in the following frames indicate the outline of the vesicle. Orange arrows indicate membrane deformations (vesicle constriction). The partially visible vesicle on the left can be seen to undergo a similar transition from a large actin network ($t = 0$) to myosin-constricted cluster ($t = 60$ min). Supplementary Movie 6 shows the same field of view. **b** Additional example of large-scale vesicle deformation through actomyosin contraction. Even though no furrow-like deformation was observed, the contraction of actin bundles still causes large-scale vesicle deformations (orange arrows).

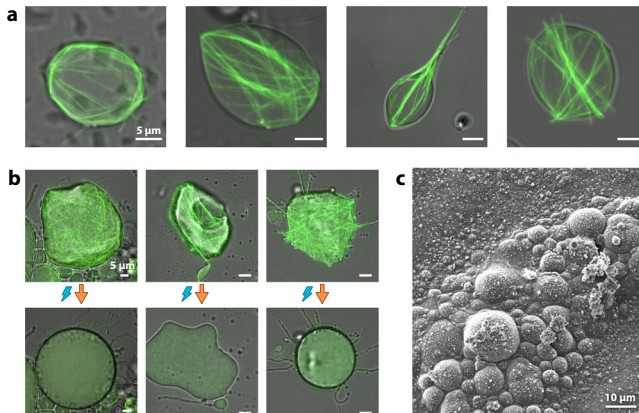

**Fig. 6 Actin network governs vesicle shape in osmotically deflated vesicles. a** Variety of vesicle shapes produced by different morphologies of encapsulated actin networks, some with stabilizing cortices and others with filopodia-like membrane protrusions. **b** Upon exposure to photodamage through increased laser power, cytoskeletal vesicles lose their stabilizing actin cortex, and often take on a flat and round shape. All three examples are shown in Supplementary Movie 7. **c** Vesicles with a stabilizing artificial actin cortex can be dried, frozen, and imaged using cryo-scanning electron microscopy.

generated and attached to vesicle membranes from the inside to transmit their contractile forces. Our experiments suggest that ring formation, membrane attachment, and contraction are not sufficient for division of these cell-like compartments, as long as the positional stability of the force-inducing machinery on the compartment surface cannot be ensured. However, toward the bottom-up design of a minimal division machinery, this system is an ideal starting point for identifying the additional parameters and components required. Furthermore, the robust and reproducible in vitro assembly methods used here provide a reliable platform for further reconstitution of the key processes of life beyond cell division.

## Methods

**Proteins.** The proteins fascin (human, recombinant), α-actinin (turkey gizzard smooth muscle), myosin II (rabbit, m. psoas), actin (alpha-actin skeletal muscle, rabbit), ATTO488-actin (alpha-actin skeletal muscle, rabbit), and biotin-actin (alpha-actin, rabbit skeletal muscle) were purchased from HYPERMOL (www.hypermol.com). VASP was expressed in *Drosophila* S2 cells, and purified using Ni-NTA affinity purification followed by gel filtration. We use the mutant vinculin$^{N773A,E775A}$ (vinculin$^{2A}$), which has reduced autoinhibitory interactions[70]. Purification of talin and vinculin$^{2A}$ was performed as described in detail in our previous publication[39]. In brief, the His-tagged proteins were expressed in *Escherichia coli* BL21 (DE3) and purified via Ni-NTA affinity purification. Talin was further purified using ion exchange purification and gel filtration chromatography on a Superdex 200 16/600 column (GE Healthcare) or Superose 6 10/300 column (GE Healthcare) in storage buffer (50 mM HEPES, pH 7.8, 150 mM KCl, 10% glycerol, and 0.1 mM EDTA). Protein purity was assessed via SDS–PAGE and gel filtration. Proteins were quick frozen and stored in aliquots at −80 °C until further use.

**Reaction mix preparation.** In all experiments, we used 10% labeled actin (ATTO488-actin). In conditions with membrane-attached actin bundles we used 4% biotinylated actin and 0.17 μM neutravidin. In all cases, actin, labeled actin, and biotinylated actin were resuspended in deionized water and pre-spun at $15,000 \times g$ for 10 min at 7 °C in a tabletop microcentrifuge. The top 75% of the actin solution was then transferred to a new Eppendorf tube and kept on ice for the duration of the experiment.

The actin concentrations we used are within the typical range used in in vitro experiments. However, it should be noted that these concentrations are much lower than the concentrations present in living cells, where a complex regulatory system controls the amount of polymerizable actin.

The reaction mix contained 10 mM imidazole, 50 mM KCl, 1 mM MgCl$_2$, 1 mM EGTA, 0.2 mM ATP, pH 7.5, and 15% iodixanol (from OptiPrep™, Sigma Aldrich). For the experiments shown in Fig. 6 and Supplementary Movie 7, the solution surrounding the GUVs contained 10 mM imidazole, mM KCl, 1 mM MgCl$_2$, 1 mM EGTA, 0.2 mM ATP, pH 7.5, and 200 mM glucose. For the deflated GUVs in Fig. 6 and Supplementary Movie 7 the outer solution contained a higher glucose concentration (300 mM).

For the experiment in Fig. 5, we used 0.1 μM myosin II.

**Lipids.** The preparation of the lipid-in-oil mixture is based on published protocols[27,71]. We use POPC (1-palmitoyl-2-oleoyl-sn-glycero-3-phosphocholine, Avanti Polar Lipids, Inc.) with 1% DSPE-PEG(2000) biotin (1,2-distearoyl-sn-glycero-3-phosphoethanolamine-N-[biotinyl(polyethyleneglycol)-2000], Avanti Polar Lipids, Inc.; both 25 mg/ml in chloroform) and give 77 μl thereof in a 10 ml vial with 600 μl chloroform. In experiments with labeled membranes, 3 μl DOPE-ATTO655 (0.1 mg/ml in chloroform) is added. While being mixed on a vortex mixer, 10 ml of a silicon oil and mineral oil (Sigma Aldrich, M5904) mix (4:1 ratio) is slowly added to the lipid solution. Since the lipids are not fully soluble in this mix of silicon oil, mineral oil, and chloroform, the resulting liquid is cloudy.

**Vesicle generation.** Vesicles were produced using the cDICE method as described by Abkarian et al.[45] with modifications we described in a previous publication[46]: instead of utilizing petri dishes, we 3D printed the rotating chamber, in which the vesicles are generated. Inner diameter of chamber: 7 cm, diameter top opening: 3 cm, height of chamber: 2 mm. (CAD file available at https://doi.org/10.5281/zenodo.4555840.) Printed with Clear Resin on Formlabs Form 2. A magnetic stirring device (outdated IKA-COMBIMAG RCH) served as a motor, after the heating unit was removed to expose the motor shaft.

The inner solution is loaded into a syringe (BD Luer-Lock™ 1-ml syringe), which is then placed into a syringe pump system (neMESYS base 120 with neMESYS 290 N) and connected through tubing to a glass capillary (100 μm inner diameter).

A total of 700 μl of the outer solution is pipetted into the rotating chamber, followed by ~5 ml of the lipid-in-oil mixture. The capillary tip is then immersed in

the oil phase and the inner phase injected at a flow rate of 50 μl/h for 10 min. The vesicles are withdrawn from the cDICE chamber with a micropipette.

The concentration of the encapsulated protein varies within a certain range. In experiments in which we encapsulated a simple soluble fluorescent dye, we found that the concentrations within the vesicle population follows a log normal distribution (Supplementary Fig. 4b). We assume that this effect is reduced for a reaction mix containing actin in the process of polymerizing and bundling. During vesicle generation, this vesicle content is much less diffusive and therefore less likely to leave the vesicles.

**Imaging**. The vesicles are pipetted into a microtiter plate for imaging (Greiner Bio-One, 384-well glass bottom SensoPlate™), each well passivated beforehand with 50 μl of 5 mg/ml β-casein (Sigma Aldrich) for 20 min.

Imaging is then performed with an LSM 780/CC3 confocal microscope (Carl Zeiss, Germany) equipped with a C-Apochromat, 40×/1.2 W objective. We use PMT detectors (integration mode) to detect fluorescence emission (excitation at 488 nm for ATTO488) and record confocal images.

Z-stack datasets of vesicles contain between 40 and 65 confocal slices (depending on vesicle size) with a slice interval of 0.5 μm with the exception of time series (Figs. 5 and 6b, and Supplementary Figs. 13 and 14), which contain less slices with a larger z interval.

**Image analysis**. Image processing and analysis is mostly performed using the software ImageJ/Fiji[72,73] and SOAX[74,75]. All images shown in the manuscript are maximum projections of z-stacks of confocal images (see Supplementary Fig. 1). Only exceptions are the images in Supplementary Figs. 2 and 3, which show single confocal images in order to highlight membrane binding. The three-dimensional representations in Supplementary Movies 1–4 are created with the Fiji command "3D Project".

For the computational, three-dimensional characterization of the actin networks, we generate skeletonized models from selected vesicles in our confocal z-stacks. The networks are identified and extracted with SOAX by active contour methods. In order to optimize the images for the identification of the filaments, the stacks are first deconvolved using the software Huygens (Scientific Volume Imaging) and then further preprocessed using Fiji. Bundle curvature (Fig. 2b) is estimated with SOAX. Membrane proximity (Fig. 3c, d) is calculated with a custom ImageJ script, which is available at https://doi.org/10.5281/zenodo.4555840.

Visualizations of the skeletonized models of actin networks by SOAX, as shown in Fig. 1d and Supplementary Movie 2, are generated in UCSF Chimera[76].

Vesicle sizes were manually measured using a MATLAB (MathWorks) script.

For more details regarding image processing and analysis see Supplementary Method.

**Statistics and reproducibility**. Most of the images show particular features that were not reproduced with identical protein concentrations, however, were reproduced under similar conditions and in sum reflect on robust underlying mechanisms. In total, we performed ≥30 independent experiments with fascin, ≥15 independent experiments with α-actinin, ≥10 independent experiments with talin + vinculin and three independent experiments with VASP that all showed similar actin morphologies, as evidenced by the respective images. All images in Fig. 1 were reproduced at least three times with similar concentrations. Results in Fig. 2 were obtained on four different vesicles for each condition within two experimental runs. Conditions as in Fig. 3 were varied once with a total of 20 conditions and incremental differences between the parameters, indicates a high degree of reproducibility. Experiments in Fig. 4 were performed two (fascin) or three (α-actinin and talin + vinculin) times. Reproducibility of experiments with myosin was poor, as described in the paper. Figure 5a shows the most "furrow-like" membrane deformation we observed. We captured time series of membrane deformation in four additional cases. Membrane deformations as shown in Fig. 6a were observed on many occasions in ≥5 experimental runs. Result shown in Fig. 6b was repeated many times within ≥3 experimental runs. Cryo-EM shown in Fig. 6c was repeated once with similar results. Experiments shown in Supplementary Figs. 1–4 were performed once. Images in Supplementary Fig. 6 are from three different experimental runs per condition. Images shown in Supplementary Figs. 8–10 were reproduced at least once with similar concentrations. Images in Supplementary Figs. 13–15 were not reproduced.

**Reporting summary**. Further information on research design is available in the Nature Research Reporting Summary linked to this article.

## Data availability
Data supporting the findings of this manuscript are available from the corresponding author upon reasonable request. A reporting summary for this article is available as a Supplementary Information file.

## Code availability
The custom ImageJ script for the membrane proximity analysis (Fig. 3c, d) is provided at https://doi.org/10.5281/zenodo.4555840.

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

## Acknowledgements
We thank Allen P. Liu for helpful discussions. We are grateful to Gunnar Goetz for experimental contributions, to David Rutkowski for contributing code to improve simulation efficiency, and Julia Skrapits for help with simulations. We thank Giovanni Cardone and Martin Spitaler from the MPI-B Imaging Facility for assistance with data analysis. This work is part of the MaxSynBio consortium, which is jointly funded by the Federal Ministry of Education and Research of Germany and the Max Planck Society. C.F.K. is a recipient of the Humboldt Research Fellowship for Postdoctoral Researchers and has received funding from the European Union's Horizon 2020 research and innovation program under the Marie Sklodowska-Curie grant agreement No 794162. N.M. acknowledges the Boehringer Ingelheim Foundation Plus 3 Program, and the European Research Council (ERC-CoG, 724209). D.V., D.H., and M.A.K. were supported by National Institutes of Health Grant R01GM114201 and R35GM136372. Use of the high-performance computing capabilities of the Extreme Science and Engineering Discovery Environment (XSEDE), which is supported by the National Science Foundation, project no. TG-MCB180021 is also gratefully acknowledged.

## Author contributions
T.L. and P.S. designed the experiments. T.L. performed experiments and analyzed the results. D.H. and M.A.K. performed computer simulations. L.B. generated electron micrographs. C.F.K. purified proteins. C.F.K. and S.K.V. performed preliminary experiments. T.L. wrote the manuscript. P.S., C.F.K., and D.V. edited the manuscript. P.S., D.V., and N.M. provided resources.

## Funding

## Competing interests
The authors declare no competing interests.
