## [Peer Review File · Nature Communications]

Reviewer #1 (Remarks to the Author):

The manuscript by Litschel et al. describes the reconstitution of actin rings inside giant unilamellar lipid vesicles (GUVs). The work falls into the growing and highly active field of bottom-up synthetic biology. Minimal actomyosin cortices have already been reconstituted into GUVs (in fact also using cDICE method, e.g. Maan et al., Adhesion of Active Cytoskeletal Vesicles, *Biophys. J.* 2018 [Ref. 28]) and actomyosin rings were previously observed in droplets (e.g. Miyazaki et al., Cell-sized spherical confinement induces the spontaneous formation of contractile actomyosin rings in vitro, *Nature Cell Biology* 2015 [Ref. 52]). Notably, contractile actomyosin networks in GUVs have already been demonstrated in a publication that was not referenced, possibly by mistake (Tsai et al., Encapsulation of Active Cytoskeletal Protein Networks in Cell-Sized Liposomes, *Langmuir*, 2011, <https://pubs.acs.org/doi/pdf/10.1021/la201604z>). However, the reproducible reconstitution of rings in GUVs is new. At the moment, this aspect is presented in Figure 2A-C. Therefore I am torn about the manuscript: It presents an interesting observation, but lacks novelty in large parts. The main aspect of novelty deserves at least an increased focus in a thoroughly revised version of the manuscript.

Major comments:

- 1) Which role do the lipids play for the membrane-anchoring of the actin? Have the authors thought about using e.g. positively charged lipids to introduce charge-mediated interactions with the periphery? Does the type of linker matter for the ring formation or merely the amount?
- 2) The authors mention that "Comparable ring formation has been shown by Miyazaki et al., who assembled actin into rings by depletion forces through the crowding agent methylcellulose, while confined in small water-in-oil droplets (Ref. 53)". This is true. Why did they then not try methylcellulose first? Did it not work? It would be very interesting to see a direct comparison between the ring assemblies in droplets and GUVs.
- 3) The statistical analysis is not satisfactory. How many vesicles were analyzed in Fig. 2A,B? What do the error bars correspond to? The same information is missing for Figure 3B and Figure 4.
- 4) In Figure 2B, the authors compare vesicles with a radius of 2.5 μm and 5 μm . This is hardly comparable with the bulk of the GUVs with diameters over 10 μm .
- 5) In Figure 2D, are the time scales of the contraction consistent for different vesicles? Is the duration until full contraction comparable to the time scale on SLBs at such low myosin concentrations?
- 6) The observation in Figure 2D have previously been described by Tsai et al. (Encapsulation of Active Cytoskeletal Protein Networks in Cell-Sized Liposomes, *Langmuir*, 2011, <https://pubs.acs.org/doi/pdf/10.1021/la201604z>). This should at least be referenced. Furthermore the differences should be discussed.
- 7) In the section "Membrane attachment shapes actin organization by curvature induction" the authors state that "Our results indicate that if bundles are sufficiently long, their confinement by the vesicle boundary forces them to bend and concentrate at the inner surface" This is no new insight. The authors should at least compare and contrast their findings to the previous results.
- 8) In Figure 4, the theoretical results may represent the acquired data but for smaller GUV sizes. Can the authors comment on this?
- 9) In Figure 5C, there is no control shown nor any other images proving the statements. How often was this tried, repeated?
- 10) In the discussion the authors state that "We have shown, as a proof of principle, that myosin mediated constriction of such ring-like membrane-bound actin structures can be induced in lipid membrane vesicles. The vesicle deformations (Figure 2D) demonstrate the strength of the

membrane anchoring and the final contracted state resembles myosin-induced symmetry breaking observed in cellular actomyosin systems under confinement.” This has been done by Tsai et al. (<https://pubs.acs.org/doi/pdf/10.1021/la201604z>, not referenced). The authors themselves state the actin is not really assembled as a ring in presence of myosin (see according section). I guess therefore they call it "ring-like", which is not really true looking at 2D for example.

11)The authors cite [Kelley et al., in revision] quite prominently. It seems to be a manuscript from the same group, which is closely related to the present work. In order to evaluate the present manuscript, I would need access to this reference. The authors should provide it as Supplementary Material for reviewers. Otherwise it is not possible to provide a fair judgement of the present work.

12)“Interestingly, while α -actinin, talin/vinculin, and VASP all produced similar morphologies, fascin bundles take on the most unique appearance. These bundles often bend in kinks when their path is obstructed by the membrane, while the other proteins form smoothly curved bundles that can follow the curvature of the encapsulating membrane (Figure 1C,D).” This statement deserves quantification. What is the persistence length? How is it different in bulk and in GUVs?

13)I wonder why other researchers have so far not reported actin rings in GUVs, although the reconstitution of actomyosin and its attachment to the membrane was successful. Can the authors comment on this? To me it seems that the adaptations of the cDICE method were crucial, in particular for the reproducibility? It would be important to describe these adaptations in more detail. In the light of open science, I also strongly recommend sharing e.g. the design file for the cDICE chamber. Only this way, it will be possible to reproduce the results in other laboratories.

Minor comments:

1)Since the authors also work on the FTSZ/Min system as a potential route to synthetic cell division, do they really believe that actomyosin-based division is easier to achieve as they state in the abstract? This could be an interesting point of discussion for the conclusion.

2)In Figure 1B and 2C, the vesicles do not appear to be spherical. Can the authors comment on this? It would be interesting to show an overlay with the brightfield image.

3)In Figure 4C/D, how is membrane proximity defined?

4)Figure 5B - Scale bars are missing for the first two GUVs.

5)How do the actin concentrations used in the GUVs compare to actin concentrations in living cells? How about the other components?

6)In the skeleton model (Figure 1D) what do the different colours represent? Do they contain z-information or do they just depict the individual bundles? How do the authors decide whether a filament ends at a branch or not (some filaments seem to continue beyond their branching points, some don't)?

Reviewer #2 (Remarks to the Author):

In this work, Litschel and coworkers successfully encapsulate actin and crosslinkers in GUVs, and demonstrate formation of rings, a strong advance towards an in vitro reconstitution of cell division. Moreover, upon inclusion of motors, they observe ring contraction, although not vesicle contraction. Particular aspects of this study are supported by fairly representative simulation studies. Overall, this work is appropriate for publication in Nature Communications. Below, I list some questions and suggestions which I feel could help clarify parts of the manuscript.

Glen Hocky
New York University

- In figure 2B, it's not clear what metric is used to define 'ring-like' structures. Given the ambiguity in this definition, it appears that you could read this plot as saying smaller or larger vesicles have more rings formed. This quantity should be precisely defined. But it would also be better if this definition could be one that can be directly compared to the experimental case. For example, fraction of bundles spanning more than X % of the circumference of the sphere.
- In a related point to the previous, follow up experiments in figure S6 support the idea that smaller vesicles have more rings. It is not fully described in the SI, but my presumption is this is just a histogram over current conditions, meaning the average actin length in each sized vesicle should be the same. This is not exactly comparable to the simulations in 2B. It would be better if the simulations in 2B could be further supported by a scan over radius at fixed L. Perhaps $L=6$, $R=4,5,6,7$.
- In figure 2D - what crosslinker is used?
- In related figure S4, are full fields of view available for the ones not currently shown? It's unclear what the distribution of shapes looks like for VASP and fascin, as compared to talin+vinculin (shown main text), and alpha-actinin (S4)
- In figure 3, the authors show a distribution of curvatures for fascin bound bundles, normalized to the spherical radius. As pointed out, the distribution is very wide. The right side of the distribution confuses me a little bit. Could the authors show examples of curvatures ~ 2 ? What is causing this curvature if not the membrane, is this what would be expected from normal fluctuations? Are these large curvature situations coming from small or large vesicles? Are the very large curvature situations kinks such as shown in A?
- In figure 4, there is a sharp jump in behavior for non-membrane bound bundles from being distributed in the center to being distributed around the 'cortex'. I wonder if this could be related to the size distribution of vesicles? Could this data be broken down a bit more? Perhaps something like average distance from the membrane vs $\text{bundle_length}/\text{radius}$ for the 6.7 and 10% case might show interesting underlying behavior (eg a transition at $\text{bundle_length}/\text{radius} \sim 1$ or $\text{bundle_persistence_length}/\text{radius} \sim 1$? - actually I suspect the latter quantity is much bigger than 1, so I don't have a prediction of where the transition would occur)
- In figure 5 and related discussion, the authors state that stabilizing actin cortices can protect vesicles from collapse. It was not clear to me what is used to define a cortex. Does a full shell of crosslinked actin have to be in place or would a more ring-like structure be sufficient? Under what conditions can cryoEM data be and not be obtained?
- In the final section, the authors discuss the slippage of rings into clusters, and note that this occurs in yeast when the cell wall is removed. I wonder if another difference could be due to the polarity of actin in the rings? In fission yeast, as famously shown by Prof. Vavylonis, contractile rings are spontaneously formed due to the action of motors in such a way that the filaments are all or predominantly anti-parallel. The crosslinker alpha-actinin is present and can support this structure.

However fascin cannot support this contractile geometry, and I'm not sure about the other two cases considered here. Regardless, can the authors describe what they expect is the polarity distribution in their rings? In simulation, can this thought experiment be performed where fully formed ring geometries are transformed into fully antiparallel structures before contraction?

- It seems that one of the impediments to testing contractility is that motors must be encapsulated with the rest of the soup of proteins, and their crosslinking and motor activity impedes formation of rings unless a small concentration is used. I speculate that future work could partially overcome this limitation using optically controlled motors. In lieu of further experiments, could the authors please discuss whether this might be achievable with current light-activatable constructs in a future experiments? (I believe I saw reported at a conference of light polymerizable myosin minifilaments, but I don't see a publication right now. That seems like it would be ideal).

Other comments -

- Are captions available for the movies? I don't seem to have them. The experimental conditions in each case should be given somewhere.

- For simulation of actin polymerization, it says in the SI that a fixed number of nuclei are created, and this is tuned to achieve a desired final length. I don't believe this initial nuclei number is listed in the paper for any of the simulation conditions. Does this number scale as a function of concentration in the cases where concentration is varied?

- The authors should re-check the supplemental details on modeling for small typographical errors. For example, in the Langevin random force equation, the left hand side should have a t' , and in the attraction between actin filaments, the second r_i should be a vector(bold), as well as those in the parentheses for the piecewise definition. Also, I believe the second order tensor $I_{\alpha\text{-}\beta}$ refers to the independence of x,y, and z directions, (α and β being axes), but this is not stated.

Reviewer #3 (Remarks to the Author):

Cell division facilitated by a contractile actomyosin ring is a crucial process for cell viability and understanding its biophysical details is critically important. Litschell et al. used synthetic biology to reconstruct ring-like actomyosin structures in giant unilamellar vesicles and provide a number of biophysical insights related to the formation, binding to the membrane and contraction of these structures. These results are confronted with a model to determine the key parameters that govern ring formation and membrane deformation.

The main new messages of this study are related to the importance of membrane binding on the probability of ring formation and the observation of membrane deformation following the constriction of the actin bundles. However, a number of concerns temper enthusiasm for the ongoing submission.

1/ I am uncomfortable with the way this manuscript is written, as it gives the impression that the scientific advances are more important than they actually are. In particular, it is astonishing that the landmark work of Miyazaki et al. (Nature Cell Biology, 2015) is not mentioned in the introduction, and only briefly referred to at the end of the discussion.

I believe that it is important to give credit to this milestone work for having reconstituted the formation of contractile actomyosin rings within liquid droplets, and for testing the significance of a large number of biophysical parameters that are important in this system. The authors should clarify unambiguously what the novelties of their study are. They should also highlight similarities and differences between both studies, particularly the relationship between compartment size and ring formation.

2/ The length of actin filaments, which is a critical parameter to evaluate for the interpretation of these experiments, is only estimated based on a number of weak assumptions such as numbers of actin nuclei and rates of nucleation. It is not satisfying, because the manuscript does not indicate either the number of nuclei (which must be, I agree, quite hard to evaluate from pre-spun actin), or the numerical values obtained from this calculation.

I recommend the authors to use an independent way to measure the length of individual actin filaments. For example, stabilization, labeling and imaging with fluorescent phalloidin would provide a much better estimate of the length of individual actin filaments assembled from these protein mixtures. In addition, as length is a critical parameter in their model, it would be critical for this study to vary the length of actin filaments under controlled conditions, for example by addition of a protein such as capping protein which would induce the polymerization of much shorter filaments. Moreover, it is not always clear at first reading whether lengths 'L' correspond to the length of individual filaments or the length of the ring filament observed inside the vesicles. Could the author please clear up for each experiment what is the estimated length of individual actin filaments, the average length of cables (or ring diameter), the expected persistence length of the cables if crosslinkers are present and the size of vesicles measured?

3/ It seems that there is a clear inhomogeneity in the amount of protein that is encapsulated in each vesicle. It is strikingly obvious in Figure S2 where neutravidin intensity varies widely between different vesicles of the same size. Could the authors provide a quantitative analysis of this variability, and tell how they took these variations into account when analyzing their results?

4/ Vesicle size seems to be an important feature for the probability of ring formation, however the authors do not give any information about the vesicle size distribution analyzed in each experiment. Also it would be important to clarify the observation of Figure S2 and S3. The volume of vesicles being proportional to R^3 and their surface to R^2 , I would expect the percentage of membrane binding to be inversely proportional to both protein concentration and vesicle size. It doesn't seem to be the case. Could the authors clarify that?

5/ The deformation of the membrane vesicle by the actomyosin ring is a key outcome of this study, however the authors do not provide any quantification of this result nor information about its reproducibility. Could the authors include for example in the caption of figure 2D, the percentage of vesicles showing this effect in each experiment, vesicle size, membrane tension, and how many times the experiment was repeated?

Minor comments

1/ Figure 2A presents the fraction of ring formation in the presence and absence of membrane binding; however, I am wondering why the data for VASP were not included in this analysis. Could the authors please include this information?

2/ The authors mention the use of POPC lipids in the text and in Figure S1 while in the material and

methods they indicate the use of DOPC

3/ There are a couple of spelling and formatting issues. For example:

- The authors write 'membraBe' instead of 'membrane' at multiple occasions
- In figure S8, the authors give reference to figure 2E instead of 2D

4/ The authors do not mention figure s7 in the main text

5/ Could the author please include in the supporting material a more detailed description of how data were analyzed? This would include a description of the bundle curvature analysis shown in figure 3B, how the length of rings is analyzed in 3D, a description of how single clearly discernible ring were identified in vesicles, etc...

6/ Figure 2B: Could the authors present the first frame also without the yellow and orange lines?

Reviewer #1 (Remarks to the Author):

The manuscript by Litschel et al. describes the reconstitution of actin rings inside giant unilamellar lipid vesicles (GUVs). The work falls into the growing and highly active field of bottom-up synthetic biology. Minimal actomyosin cortices have already been reconstituted into GUVs (in fact also using cDICE method, e.g. Maan et al., Adhesion of Active Cytoskeletal Vesicles, *Biophys. J.* 2018 [Ref. 28]) and actomyosin rings were previously observed in droplets (e.g. Miyazaki et al., Cell-sized spherical confinement induces the spontaneous formation of contractile actomyosin rings *in vitro*, *Nature Cell Biology* 2015 [Ref. 52]). Notably, contractile actomyosin networks in GUVs have already been demonstrated in a publication that was not referenced, possibly by mistake (Tsai et al., Encapsulation of Active Cytoskeletal Protein Networks in Cell-Sized Liposomes, *Langmuir*, 2011, <https://pubs.acs.org/doi/pdf/10.1021/la201604z>). However, the reproducible reconstitution of rings in GUVs is new. At the moment, this aspect is presented in Figure 2A-C. Therefore I am torn about the manuscript: It presents an interesting observation, but lacks novelty in large parts. The main aspect of novelty deserves at least an increased focus in a thoroughly revised version of the manuscript.

The referee is correct about the vast amount of work that has already been accomplished in the past years towards the reconstitution of contractile actomyosin rings in vesicles. It is true that many aspects of this task have been reported previously (thanks for reminding us of the Tsai et al. paper, which we of course knew, but missed to cite), but like always, the steps towards success become ever more incremental the longer an endeavor takes, and the more technical proficiency has been established. What we believe speaks in favor of publishing our success of reconstituting contractile ring-like structures in vesicles that are able to transform the membrane from within, is not only the beauty of the data, but also the clear work assignment for the next practical steps following from it. Very obviously, assembling contractile rings in vesicles without a machinery to position and fix these rings at defined spatial positions will not lead to success of controlled division. We thus believe that the very active and ever-growing community of bottom-up synthetic biologists will conceive this paper as a very valuable and worth to remember milestone towards a common goal, justifying its publication in *Nature Communications*.

Major comments:

1) Which role do the lipids play for the membrane-anchoring of the actin? Have the authors thought about using e.g. positively charged lipids to introduce charge-mediated interactions with the periphery? Does the type of linker matter for the ring formation or merely the amount?

In this work we investigated the effect of binding bundled actin to the inside of membranous giant vesicles. We chose to bind actin through specific interactions via biotin-neutravidin linkers, as this can be considered to mimic the specific interaction of the cytoskeleton with membranes *in vivo*. Cells rely on specific interactions, i.e. binding of actin through specific linker proteins. While nonspecific interactions of actin with membranes are subject of current studies (see recent paper by Schroer et al, DOI:0.1073/pnas.1914884117), it is generally thought that these are not relevant *in vivo*. Most *in vitro* studies that combine actin and membranes utilize specific interactions mediated by biotin-neutravidin bonds, or for example histidine-nickel-NTA links. We did, however, not compare different membrane linkers in our study. Here we focused on and tested several concentrations of biotinylated lipid and explored a range of biotinylated actin. While we agree that it would be interesting to test other mechanisms of membrane binding in detail, most of them require several components (e.g. in our biotin-neutravidin-biotin system, 3 components) which tremendously complexify matters. For a meaningful

comparison, the parameter landscape would be significant, without leading to conceptually new insights with regard to a minimal system. While in future studies it will be interesting to look into nonspecific charge-mediated binding, we think that the complexity of these studies might distract from the message of our paper.

2)The authors mention that “Comparable ring formation has been shown by Miyazaki et al., who assembled actin into rings by depletion forces through the crowding agent methylcellulose, while confined in small water-in-oil droplets (Ref. 53)”. This is true. Why did they then not try methylcellulose first? Did it not work? It would be very interesting to see a direct comparison between the ring assemblies in droplets and GUVs.

Similar to point 1), we tried to focus on specific interactions between biomolecules by using physiological cross-linking proteins and cell-mimicking lipid bilayer vesicles. A central theme in our paper is to emphasize the binding of bundles to the vesicle membrane to promote condensation into contractile rings able to induce membrane deformations. It is hard to compare this aspect of membrane binding between droplets and vesicles. Miyazaki et al did not look into membrane binding and solely used droplets as means of confinement. In our experience the lipid bilayer of a membranous vesicle is a much more robust and defined lipid structure than the monolayer of a water-in-oil droplet. We think that both the use of depletants as a means to bundle actin and the use of droplets make a direct comparison between the two experimental systems difficult.

We adjusted the text and discuss the differences between our approach and that of Miyazaki et al. in more detail in the introduction, in the results section, as well as the discussion of our manuscript, also in response to point 1 raised by Reviewer 3.

3)The statistical analysis is not satisfactory. How many vesicles were analyzed in Fig. 2A,B? What do the error bars correspond to? The same information is missing for Figure 3B and Figure 4.

We apologize for the lack of statistical information. We did forget to communicate a lot of this in manuscript and only submitted the information with the data reporting spreadsheet. We thank the reviewer for pointing out this oversight. We now include the requested information in the figure legends.

In the original submission we analyzed 1228 GUVs for Figure 2A (now Figure 4A), but we reanalyzed the figure to include only a small size range of vesicles (see point 4 by reviewer 3), which would have reduced this to a total number of 301 vesicles. However, in the revised version of the manuscript, we now include more data, so that a total of 1545 GUVs and 392 within the size range 15-20 μm are considered.

Error bars in Figure 3B (now Figure 2B) display only half of the error range, by only showing the negative range. We were unsure if a histogram plot like the one shown in the figure demands error bars, and had the impression full error bars would make the plot hard to read. Therefore we thought this compromise would be a good idea. We are however happy for suggestions.

4) In Figure 2B, the authors compare vesicles with a radius of 2.5 μm and 5 μm . This is hardly comparable with the bulk of the GUVs with diameters over 10 μm .

We added more simulations, comparing ring formation probability now with four different diameters ranging from 5 to 12 μm . While this certainly still does not represent the full range of vesicle sizes in our experiments, we now show that the general trend that ring formation is more likely for small sizes is also reflected in our simulations (Figure S12).

The amount of time required for our simulations increases rapidly with confinement size (see plot below), therefore certain sizes are unfortunately not feasible at the moment. The graph below shows the approximate number of days (in real time) it takes to run a simulation to 1,500 s as a function of confining radius with a persistence length of 17 μm , filament length of 6 μm and actin concentration of 2 μM . This is the time for an individual run using 4 cores in Java (blue diamonds: Lehigh Sol <https://confluence.cc.lehigh.edu/display/hpc/Sol>; red square: desktop with Intel Xeon E5-1620 3.5 GHz). To obtain meaningful averages, we run several of such simulations in single nodes. Similar times are required in a C++ implementation of the Brownian dynamics method (improvement by a factor close to 2 with the same number of threads), which was not used in this manuscript. It was unfortunately not feasible to run simulations with confinement diameters larger than 12 μm with our current methods.

5) In Figure 2D, are the time scales of the contraction consistent for different vesicles? Is the duration until full contraction comparable to the time scale on SLBs at such low myosin concentrations?

Our work differs fundamentally from any work involving actomyosin we are aware of, in that we use myosin on heavily bundled actin. This, we believe, is also responsible for different actomyosin dynamics. Our impression is that with the concentrations we use, bundling outweighs contraction initially and (other than bundles becoming thicker) there are no dynamic processes visible in the beginning. Over time however, a threshold is reached after which the network collapses. Luckily this late onset of contraction is desirable in our experiments, as we cannot image the vesicles for ~ 15 minutes after vesicle formation, which we further aid by using low concentrations of myosin. Below we include a quantification of both the onset of contraction as well as the duration from first visible contraction to a fully collapsed state.

In our experiments the onset of visible contraction between vesicles can significantly differ between vesicles. While in a few cases the actin contracted right when we started imaging the vesicles (~15 min after vesicle preparation) or even before, most contractions started after about one hour after. There seems to be a slight correlation between vesicle size and the speed of the contraction, as contractions in larger vesicles seem to take longer.

6)The observation in Figure 2D have previously been described by Tsai et al. (Encapsulation of Active Cytoskeletal Protein Networks in Cell-Sized Liposomes, Langmuir, 2011, <https://pubs.acs.org/doi/pdf/10.1021/la201604z>). This should at least be referenced. Furthermore the differences should be discussed.

We apologize for not referencing this work, as it was a valuable contribution to the field, not only regarding the major methodological advances at the time, but also from a reconstitution perspective.

We added three references to the paper and added text in the introduction and in the discussion mentioning their work. The two sentences read:

Introduction: "While, Tsai et al, showed the reconstitution of a contractile network in vesicles,²⁶ and others have reconstituted actomyosin networks in vesicles that imitate actin cortices^{27,29}, contractile actomyosin rings have proven difficult to achieve."

Discussion: "For example, Tsai et al. encapsulated a contractile actomyosin system in vesicles that condensed into dense clusters.²⁶"

We agree that the final state in our experiments resembles what Tsai et al. 2011 show in their vesicles after myosin contraction leading to the formation of an actomyosin cluster and thus symmetry breaking.

Since this is the main concern in point 10) by reviewer 1, here we would like to highlight other differences to our experiments.

We assume when stating that the results in our Figure 2D (now Figure 5A) have been shown in the mentioned paper, the reviewer is referring to Figures 6E,F in the paper (see below), as these are the only results shown with contractile actomyosin in vesicles.

Tsai et al.,
Encapsulation of Active Cytoskeletal Protein
Networks in Cell-Sized Liposomes
Langmuir 2011, 27, 10061–10071

These images show an actomyosin network in its contracted state, however do not show the dynamic process of the contraction and also do not show membrane deformations (nor do the authors claim that membrane deformations took place in the process).

One major achievement of our work is the observation of membrane deformations through myosin contractions, which Tsai et al. did not report. We think that this difference between our work and theirs nicely demonstrates that higher order actin organization (in the form of bundling) is important for locally inducing force that then leads to membrane deformations, like here through the constriction of ring-like structures.

7) In the section “Membrane attachment shapes actin organization by curvature induction” the authors state that “Our results indicate that if bundles are sufficiently long, their confinement by the vesicle boundary forces them to bend and concentrate at the inner surface” This is no new insight. The authors should at least compare and contrast their findings to the previous results.

We agree that the underlying physical principles are rather simple and we were careful to not display this finding as novel and a first-time observation. Namely, the explanation for this effect likely has to do with long bundles/filaments minimizing bending energy by assuming the smallest possible curvature. Unfortunately, we did not find any reports in which a similar behavior (formation of a cortex-like structure) is seen for bundled actin or other stiff filaments. However, Tsai et al., 2015 mentions that in some cases they saw unbundled actin filaments to be more concentrated closer to the membrane, so we mention this now in our manuscript. If the reviewer is aware of findings more similar to ours, we would be grateful for any references.

We assume, however, that the same physical principle (minimization of bending energy) is responsible for the formation of single rings in in our vesicles without membrane attachment. This effect of ring formation in spherical confinement was observed by Miyazaki et al (Nature Cell Biology, 2015) in droplets. We now

(also in reply to point one by reviewer 3) discuss this paper in more detail in our introduction, results section and discussion, and among other points, for example with descriptions like the in following:

“They [Miyazaki et al.] showed not only that the formation of equatorial rings from actin bundles is a spontaneous process that occurs in spherical confinement in order to minimize the elastic energy of the bundles, [...].”

8)In Figure 4, the theoretical results may represent the acquired data but for smaller GUV sizes. Can the authors comment on this?

This is indeed the case and in the new manuscript we have added experimental and simulation analysis as a function of size in Figure S12 as mentioned above, for actin concentration $c = 2 \mu\text{M}$. The $2 \mu\text{M}$ simulations do represent the behavior in the experiments for smaller GUV sizes.

Former Figure 4 (now Figure 3) also showed simulation results for $c = 6 \mu\text{M}$, for which we had to use smaller confining diameter and could not perform a systematic change of diameter. We mention this in the discussion of Figure 3 in the main text.

9)In Figure 5C, there is no control shown nor any other images proving the statements. How often was this tried, repeated?

We intended to show this as a qualitative proof of concept. The experiment was only done twice and only one experimental run resulted in electron micrographs. We did not perform any quantitative analysis and we tried to convey the results of the control (without a cytoskeletal cortex) in our supplements by showing an image of very deformed vesicles. However, we know from previous experiments that vesicles without a stabilizing cortex do not survive cryo-fixation and it was rather surprising to see that these “cytoskeletal vesicles” survive this process.

We adjusted the text and stressed more that this experiment should be seen as a proof of concept. If the reviewer still perceives the results as misleading, we would not hesitate to remove the image in Figure 5C (now Figure 6C) entirely from the manuscript, as we think it is not a major or important part of our work, but rather an unexpected curiosity that is not necessarily needed.

10)In the discussion the authors state that "We have shown, as a proof of principle, that myosin mediated constriction of such ring-like membrane-bound actin structures can be induced in lipid membrane vesicles. The vesicle deformations (Figure 2D) demonstrate the strength of the membrane anchoring and the final contracted state resembles myosin-induced symmetry breaking observed in cellular actomyosin systems under confinement." This has been done by Tsai et al. (<https://pubs.acs.org/doi/pdf/10.1021/la201604z>, not referenced). The authors themselves state the actin is not really assembled as a ring in presence of myosin (see according section). I guess therefore they call it "ring-like", which is not really true looking at 2D for example.

While Tsai et al. 2011 neither looked at bundled actin, nor see any membrane deformations or show dynamic processes (as discussed above), we assume the reviewer is referring to the final state, which is the results of a symmetry breaking event. We agree that the results Tsai et al 2011 show in their Figure 6E,F (see above) resemble our experiment after the ring-like network contracts into one point. In other studies (Carvalho et al. 2013, Loiseau et al. 2016, Ierushalmi et al. 2020) similar symmetry breaking events through myosin contraction have been seen, as was referenced by us at the end of the second sentence the reviewer quoted above. We apologize that we missed citing this earlier study (Tsai et al. 2011) that reported actomyosin symmetry breaking in confinement. We now reference Tsai et al. 2011 along with the other references and mention their work separately in the discussion: *““For example, Tsai et al. encapsulated a contractile actomyosin system in vesicles that condensed into large clusters.””*

As the reviewer points out, we mention in the manuscript that the inclusion of myosin in the vesicles hampers the formation of single actin rings. The example we show in Figure 2D (now Figure 5A) was the most ring-like actin network we observed contracting. Unfortunately, in this specific case, our imaging parameters were not set ideally, and we did not record the full 3D volume of the vesicle, making it difficult to see the ring-like character. We now clearly state when presenting Figure 2D (now Figure 5A), that the figure example shows the only comparable instance of such an event. We now also point out we see this example as a proof of principle, and an indication of what can be achieved when combining actin bundling, membrane anchoring and myosin contraction. While at this stage, myosin contraction hampers experimental consistency, future studies will likely explore how to specifically induce this activity, introducing more control over this experimental setup.

We agree that from the 2D projection the “ring-like” character in Figure 2D (now Figure 5A) was not obvious, therefore we now added a supplementary figure (Figure S11) with 3D information of the images, i.e. showing the z-stack reconstruction from a 90° angle (“side view”). The missing top and bottom z-slices are more obvious in this representation, however by carefully looking at the dataset in its entirety we think that we can say with certainty that the bundles connect into one continuous ring as depicted in Figure S11D.

11)The authors cite [Kelley et al., in revision] quite prominently. It seems to be a manuscript from the same group, which is closely related to the present work. In order to evaluate the present manuscript, I would need access to this reference. The authors should provide it as Supplementary Material for reviewers. Otherwise it is not possible to provide a fair judgement of the present work.

We attached the requested manuscript “Phosphoinositides regulate force-independent interactions between talin, vinculin, and actin” by C. Kelley, T. Litschel, S Schumacher, D. Dedden, P. Schwille and N. Mizuno with the revisions. The paper is now published in eLife (DOI:10.7554/eLife.56110). The paper demonstrates the bundling capability of talin and vinculin and their dependency on PIP2. The last figure further shows experiments in which encapsulation in PIP2-rich vesicles is used to demonstrate the interactions of PIP2 with talin and wildtype vinculin. (In our paper we use a vinculin mutant (also described in Kelley et al.) which does not require PIP2 for its bundling activity.)

12)“Interestingly, while α -actinin, talin/vinculin, and VASP all produced similar morphologies, fascin bundles take on the most unique appearance. These bundles often bend in kinks when their path is obstructed by the membrane, while the other proteins form smoothly curved bundles that can follow the curvature of the encapsulating membrane (Figure 1C,D).” This statement deserves quantification. What is the persistence length? How is it different in bulk and in GUVs?

In Figure 1 we solely wanted to demonstrate the robustness and variability of the system and not to make any novel claims about properties of bundling proteins. We admit that this is not apparent from the statement in the manuscript and the text was quite misleading (making it appear like a novel finding). Among other small changes, we now deleted the first part in the following sentence ~~“To determine whether different bundling proteins could achieve unique higher order actin networks, we tested four different types of actin bundling proteins”~~.

Moreover, the higher bending stiffness of fascin was described in previous publications and we should have made this more clear. We now refer to existing research when describing our figure 1. However, if the referee thinks an omission of this section would be better, we would simply adjust the manuscript so that there is no description of the different phenotypes of bundling between the bundlers.

We judged that a quantitative comparison of the different bundlers with or without confinement would require work that goes beyond the scope of this paper – not only because these properties were already investigated in other work in bulk and in droplets, which both allow for more control and thus are better approaches to quantify bundle parameters. The analysis of persistence length in confinement under mechanical deformation and kinks is not straightforward. Further, a meaningful quantitative comparison would involve testing a large range of different concentrations of each bundling protein, especially as we use different concentrations for different bundling proteins by default (see figure 1 caption). These different concentrations are necessary, as for example talin with vinculin (each 2 μ M) barely form any bundles when we use them at the same concentrations as we use fascin (0.3 μ M). We think this aspect makes a quantitative comparison difficult as persistence length and other parameters will depend on bundling protein concentration. As such, these experiments would require covering a large range of concentrations for each bundling protein to result in any meaningful statement. We think it goes beyond the scope of this paper to perform such experiments and would most likely distract from the main messages, which is not the comparison of different bundling proteins.

13)I wonder why other researchers have so far not reported actin rings in GUVs, although the reconstitution of actomyosin and its attachment to the membrane was successful. Can the authors comment on this? To me it seems that the adaptations of the cDICE method were crucial, in particular for the reproducibility? It would be important to describe these adaptations in more detail. In the light of open science, I also strongly recommend sharing e.g. the design file for the cDICE chamber. Only this way, it will be possible to reproduce the results in other laboratories.

To our knowledge, the combination of bundling actin through bundling protein and binding it to the vesicle membrane through specific interactions is rather new in the field. We are only aware of one other publication in which this has been done: Maan et al., 2018, *Biophys. J.*, 115, 2395–2402 (DOI:10.1016/j.bpj.2018.10.013). In this publication however, one and the same protein (his-tagged anillin) acts as both, actin bundler and actin-membrane linker.

We agree with the reviewer that the lack of similar results in the field is most likely a matter of reproducibility with other experimental systems. While there have been more than a dozen papers over the years that show encapsulation of actin within vesicles, we know from personal correspondence with respective authors and from our own experience with these experimental systems, that the lack of reproducibility and robustness is a persistent problem. In many of these, the low number of vesicles per experimental run and variation in experimental outcome despite seemingly identical conditions does not allow for any quantitative conclusions and hamper the encapsulation of more complicated protein systems with many components. We think that emulsion transfer methods in general, and more particularly cDICE, does not suffer from these problems that affect older vesicle generation techniques.

We added a link in our supplementary PDF that leads to the CAD design file of our cDICE chamber design.

Minor comments:

1) Since the authors also work on the FtsZ/Min system as a potential route to synthetic cell division, do they really believe that actomyosin-based division is easier to achieve as they state in the abstract? This could be an interesting point of discussion for the conclusion.

We thank the reviewer for this suggestion. Our understanding is that this reply letter will be published online, therefore we think this reply letter might be a good platform to address this in detail.

There are several reasons why we think that actomyosin may be a better platform.

- Firstly, Z-rings (composed of FtsZ) are intrinsically small. Bacterial cells are only about 1 micron in size and so are Z-rings that have reconstituted *in vitro*, even when reconstituted on flat membranes. Therefore, artificial vesicles potentially divided by a Z-ring would hardly be GUVs, by definition of the term “GUV” (diameters between 1 μm and 200 μm). The size of GUVs is one of their major advantages for reconstitution experiments, as it allows the observation of processes in detail with sufficient resolution using fluorescence light microscopy. While reconstitution of FtsZ in larger vesicles has led to new insights in the field, it would likely not allow for the division of a cell-shaped vesicle. While in our study we were not yet able to show a full and clear division of a spherical vesicle, we think that actomyosin rings are so far the best candidates to enable exactly that.

-The mechanism by which FtsZ constricts bacterial cells is not known, and it is still debated whether the force for constriction could be exerted by FtsZ itself. So far *in vitro* reconstitution by our group and others has not convincingly shown that forces sufficiently high for cell division can be produced.

-Lastly, actomyosin-induced vesicle contraction might be relevant for a broader audience, since it concerns eukaryotic cells, while FtsZ-mediated division is restricted to a certain number of bacterial cells (and mitochondria).

2) In Figure 1B and 2C, the vesicles do not appear to be spherical. Can the authors comment on this? It would be interesting to show an overlay with the brightfield image.

We now include phase contrast images (differential interference contrast) in the supplement. Most vesicles appear spherical in their x-y cross section (except for when they are in too close contact with other vesicles, which can flatten the contact area) and also have almost the same diameter in z, thus are almost spherical.

We are wondering if this misconception might be due to vesicles like the two vesicles marked below, which might appear elliptical in the projection view image. This is actually a quite interesting reoccurring phenomenon. We often see that actin networks assemble into planar structures, as can be seen in the attached figure (on the right). We agree that from certain angles these look like they are encapsulated in an ellipsoid vesicle.

3) In Figure 4C/D, how is membrane proximity defined?

We apologize for generally failing to provide a detailed description of our analysis methods. We now describe our analysis at the end of the manuscript and in much more detail in our supplementary information. We now also include the ImageJ Macro script file as a supplementary file.

4) Figure 5B - Scale bars are missing for the first two GUVs.

We apologize for that, we added these scale bars for these figure panels.

5) How do the actin concentrations used in the GUVs compare to actin concentrations in living cells? How about the other components?

In vivo, actin is an extremely tightly regulated protein and a large percentage of the protein is kept under conditions under which it cannot polymerize, e.g. by monomers being bound to proteins such as profilin. Typically for *in vitro* experiments, concentrations orders of magnitude smaller compared to in cells are used. In the early days of *in vitro* reconstitution, actin concentrations more similar to those in cells were frequently used. At these concentrations (50 to 200 μM), unregulated, polymerized actin forms crystalline structures (e.g. "liquid crystals"), which, to our knowledge, have not been identified to be of importance *in vivo*. In our study, we chose concentrations that are well within the usual range used in most current *in vitro* experiments. We now include the following in the methods section in the supplementary information:

"The actin concentrations we used are within the typical range used in in vitro experiments. However, it should be noted that these concentrations are much lower than the concentrations present in living cells, where a complex regulatory system controls the amount of polymerizable actin."

6) In the skeleton model (Figure 1D) what do the different colours represent? Do they contain z-information or do they just depict the individual bundles? How do the authors decide whether a filament ends at a branch or not (some filaments seem to continue beyond their branching points, some don't)?

In our representation different colors represent what the algorithm defines as different bundles. Our algorithm automatically detects linear bundles without furcations (and also decides based on curvature whether a bundle is continuous or not). For this it does not require any manual decisions. We briefly describe this in the methods section of our paper, which we have drastically extended as part of our revisions.

This separation into different bundles by the algorithm does not necessarily represent a real state, as the actual actin bundles do fork and in many cases all visible bundles within one vesicle are physically connected. However, we think this representation helps to visually distinguish different parts of the

network, for example when bundles crisscross in the 2D projections shown in our figures. Therefore, we did not unify colors or alter them in a different way.

For z information, the reader can refer to supplementary Movie S1, as the four vesicles that are shown in Figure 1D are shown in a rotating 3D view in Movie S1, as we now also mention in the figure caption of Figure 1D.

Reviewer #2 (Remarks to the Author):

In this work, Litschel and coworkers successfully encapsulate actin and crosslinkers in GUVs, and demonstrate formation of rings, a strong advance towards an in vitro reconstitution of cell division. Moreover, upon inclusion of motors, they observe ring contraction, although not vesicle contraction. Particular aspects of this study are supported by fairly representative simulation studies. Overall, this work is appropriate for publication in Nature Communications. Below, I list some questions and suggestions which I feel could help clarify parts of the manuscript.

**Glen Hocky
New York University**

- In figure 2B, it's not clear what metric is used to define 'ring-like' structures. Given the ambiguity in this definition, it appears that you could read this plot as saying smaller or larger vesicles have more rings formed. This quantity should be precisely defined. But it would also be better if this definition could be one that can be directly compared to the experimental case. For example, fraction of bundles spanning more than X % of the circumference of the sphere.

We now give a clearer definition in the figure caption.

For the experimental analysis of what is now Figure 4, we define “rings” as a single structure of a fully closed (no open ends) bundle with no forking points. A vesicle only counts as containing a “ring” if it does not contain any other visible bundles other than the one ring. The only “acceptable” deviations from a perfect ring is that we also count rings with kinks (e.g. for fascin) and rings that are not perfectly in a plane as “rings”.

For the simulations, we add an additional state called “ring-like” structures, since here we more often see formations that are close to what could be considered a ring, but have some imperfections like forking small bundles that stick out of the main ring. We see these less frequently in our experiments. One explanation for this discrepancy is that the formations we see in experiments would agree better with simulations with longer run-times and allowing some filament breakage when the curvature of individual filaments becomes high enough. With longer simulation times these “sticking out” bundles potentially merge with the main bundle (the ring) and thus over time a clean single ring could form, as in our experiments.

To explain this issue, we added as section “Classification of ring (R) and ring-like (RL) structures in simulations” in the supplementary text.

- In a related point to the previous, follow up experiments in figure S6 support the idea that smaller vesicles have more rings. It is not fully described in the SI, but my presumption is this is just a histogram over current conditions, meaning the average actin length in each sized vesicle should be the same. This is not exactly comparable to the simulations in 2B. It would be better if the simulations in 2B could be further supported by a scan over radius at fixed L. Perhaps L=6, R=4,5,6,7.

We agree with the reviewer and thus we performed more simulations as suggested. Supplementary Figure S12C now shows results of simulations with radii 2.5, 4, 5 and 6 μm with a 2 μM actin concentration and final filament lengths of 6 μm . Since simulations at larger radii become computationally intensive (see response to comment 4 of Reviewer 1), and since the trend is already clear, we stopped at 6 μm radius. These simulations are compared to experimental measurements as a function of vesicle size in Figure S12. The smaller vesicles do in fact have more ring structures than larger vesicles when only the confinement size is varied, in agreement with the simulation results.

- In figure 2D - what crosslinker is used?

We apologize for not clearly stating this. It is the same bundler as in Figure 2C (now Figure 4C): Talin in combination with Vinculin. We now include this information both in the figure itself and in the figure caption.

- In related figure S4, are full fields of view available for the ones not currently shown? It's unclear what the distribution of shapes looks like for VASP and fascin, as compared to talin+vinculin (shown main text), and alpha-actinin (S4)

We split the previous figure S4 into two figures, one of which focuses on rings (new Figure S6) the other one giving an overview of different morphologies for different bundlers (mostly field of view images) (new figure S7). As requested, we added a field of view image of membrane-bound fascin bundles to supplementary Figure S7. However, unfortunately we only performed few VASP experiments, as we also explain in our response to minor comment 1 of reviewer 3. We did however include more images of single vesicles in figure S6 now.

- In figure 3, the authors show a distribution of curvatures for fascin bound bundles, normalized to the spherical radius. As pointed out, the distribution is very wide. The right side of the distribution confuses me a little bit. Could the authors show examples of curvatures ~ 2 ? What is causing this curvature if not the membrane, is this what would be expected from normal fluctuations? Are these large curvature situations coming from small or large vesicles? Are the very large curvature situations kinks such as shown in A?

We only analyzed vesicles with a relatively narrow size distribution between 10 and 13 μm radius.

As most of our images show, especially bundles bundled by fascin usually do not form rings that are perfectly aligned with the membrane (curvature = 1). While many bundles are rather straight, and therefore contribute to curvatures below 1, curvatures of more than 1 also occur.

We defined curvature as $1/r$. Below we attach a schematic giving a better idea of what different curvatures ≥ 1 look like. We hope this clarifies some of the points raised by the reviewer.

Sharp kinks most likely contribute to the “More” histogram bin in our plot. However, since we divide the network into equal-length segments for our analysis (see updated methods section) and kinks are very short segments, they are not represented well in this histogram.

Longer segments with higher curvature, which the reviewer is presumably talking about, are surprisingly rare in our analysis. Segments with curvatures between 2 and 4 make up only 14% of the cases, while 83% have curvatures between 0 and 2. We also like to point out that even if a bundle is continuously attached/following the membrane, it can still bend “laterally” on the membrane, resulting in a higher curvature. But as the reviewer said, there might be some fluctuations. It is likely that our quantitative analysis does produce some systematic error that shifts the general curvature towards a higher curvature, as any inaccuracy in tracking of the actin bundles likely result in slightly more bent bundles in the generated snake model. We think our Movie S2 gives a good impression of the capabilities and inaccuracies of the 3D-snake models that we generate and that are used for further analysis.

We apologize if there was any confusion in regards to our analysis, the snake generation process, or plotting of the graph, as this information was unfortunately missing from the manuscript. We now include a detailed description of this process in the supplements.

- In figure 4, there is a sharp jump in behavior for non-membrane bound bundles from being distributed in the center to being distributed around the `cortex'. I wonder if this could be related to the size distribution of vesicles? Could this data be broken down a bit more? Perhaps something like average distance from the membrane vs bundle_length/radius for the 6.7 and 10% case might show interesting underlying behavior (eg a transition at bundle length/radius~1 or bundle_persistence_length/radius~1?- actually i suspect the latter quantity is much bigger than 1, so I don't have a prediction of where the transition would occur)

We agree with the reviewer that this is an interesting phenomenon. In the early phases of the project we wanted to highlight this effect, but later decided against that as to not confuse the reader and distract from the main points (which only emerged later in the project). One other reason this is not discussed more in the paper is because of the difficulty of quantifying bundle length. While we made several attempts to do so, we could not find a reliable way to automatically track and measure bundles in vesicles with many short bundles. Our algorithm often failed to distinguish the numerous individual bundles, resulting in clusters of bundles appearing merged in the 3D model. The problem of resolving and distinguishing the short filament bundles for these conditions is additionally hampered because more unbundled actin is in the vesicles, increasing the “background signal” significantly (see supplementary Figure S5).

We expanded our description of this effect in the manuscript and will address it in more detail in the following. We also would like to mention that Figure S6 in our supplement is partially related to this topic.

The system essentially undergoes a bundling transition towards long bundles above a certain cross-linker concentration, however the details of this transition depend on several factors, including the nature of cross-linkers and kinetics, as was also outlined in the modeling study of Adeli Koudehi et al. We generally found that approximately linear actin bundles begin to form and elongate with increasing bundling protein concentration, as can be seen in the microscopy images. Part of the “sharp jump” can be attributed to the fact that leading up to the jump, the “membrane proximity value” becomes more negative, meaning that there is more actin towards the center of the vesicle. This can be explained because straight bundles only ever ‘touch’ the curved membrane with their ends, resulting in a lower actin density right at the membrane. The longer these bundles, the more pronounced this effect becomes, until they reach a threshold (bundle length = vesicle diameter) at which all bundles meet in the center, crossing over each other. This simplified model is shown as a 2D depiction of ‘growing’ bundles in a circular confinement in the schematic below. The case that we just described is shown in the third image. A similar effect occurs in 3D (Adeli Koudehi et al and references therein).

If bundles are only slightly larger than the diameter of the vesicle, they bend and ‘free up’ space in the center of the vesicle, (where, just in the previous condition, actin was the most concentrated!) and actin density becomes high closer to the membrane (4th image).

There is, however, an additional factor, as we think that with increasing bundle length, bundling efficiency gets an additional boost. When bundles reach a length that forces them to align with the curvature of the membrane, they not only get condensed into the “2D” space close to the membrane but even lose a

degree of freedom regarding their orientation, both making them more likely to bundle. This effect can also be seen when comparing the membrane-bound condition and the non-membrane-bound conditions for 6.7% fascin:actin at 6 μ M actin (Figure S6) consistent with our numerical simulations of Figure 3. Membrane-binding seems to facilitate the formation of long bundles in this “threshold region” (6.7% fascin:actin), potentially because actin bundles are restricted to a quasi 2D-space (in spherical coordinates). Generally we can exclude size-related artifacts.

- In figure 5 and related discussion, the authors state that stabilizing actin cortices can protect vesicles from collapse. It was not clear to me what is used to define a cortex. Does a full shell of crosslinked actin have to be in place or would a more ring-like structure be sufficient? Under what conditions can cryoEM data be and not be obtained?

The vesicles shown in Figure 5C (now Figure 6C) were prepared with 6 μ M actin and high fascin to actin ratio. As can be seen in Figure 4 (now Figure 3), this does not result in the formation of rings, but rather results in an actin network that coarsely covers most of the inner membrane. Previous approaches that demonstrate the mechanical stabilization GUVs make use of a homogeneous, continuous shells. As we now more clearly state in the manuscript, we see Figure 5C (now Figure 6C) as a proof of principle, but do not intent to include any quantitative assessment of GUV survival rates. However, we can clearly say that vesicles that do not contain actin, generally do not survive the freezing process.

- In the final section, the authors discuss the slippage of rings into clusters, and note that this occurs in yeast when the cell wall is removed. I wonder if another difference could be due to the polarity of actin in the rings? In fission yeast, as famously shown by Prof. Vavylonis, contractile rings are spontaneously formed due to the action of motors in such a way that the filaments are all or predominantly antiparallel. The crosslinker alpha-actinin is present and can support this structure. However fascin cannot support this contractile geometry, and I'm not sure about the other two cases considered here. Regardless, can the authors describe what they expect is the polarity distribution in their rings? In simulation, can this thought experiment be performed where fully formed ring geometries are transformed into fully antiparallel structures before contraction?

The reviewer brings up the important issue of filament polarity in generating stable contractile rings. Indeed, myosin activity on antiparallel filaments must be maintained through the contraction process. This would require the creation of a ring consisting of antiparallel filaments as well as mechanisms to avoid contractile instabilities in which the ring may separate into non-contractile elements or collapse into a single spot. The reviewer is correct in that fascin, which should lead to bundles consisting mostly of parallel actin filaments, may not be ideal for contractile rings. Our experiments in this paper with myosin contraction used talin and vinculin as cross-linkers, which should lead to bundles of antiparallel filaments. In our simulation we did not distinguish between filament polarity during cross-linking so our simulated bundles have antiparallel filaments. However, several modeling papers have addressed the issue of polarity and polarity sorting in contractile actomyosin bundles and turnover of actin and myosin seems to be important for stability in cytokinesis. Indeed in Bidone et al. Biophys. J. 2014, we proposed that such turnover may explain how cable-like bundles of formin-generated parallel filaments transform into

antiparallel bundles). Turnover of components may be another key issue that further cell division reconstitution experiments may need to address. We have added comments in the Discussion that hopefully clarify these issues.

- It seems that one of the impediments to testing contractility is that motors must be encapsulated with the rest of the soup of proteins, and their crosslinking and motor activity impedes formation of rings unless a small concentration is used. I speculate that future work could partially overcome this limitation using optically controlled motors. In lieu of further experiments, could the authors please discuss whether this might be achievable with current light-activatable constructs in a future experiments? (I believe I saw reported at a conference of light polymerizable myosin minifilaments, but I don't see a publication right now. That seems like it would be ideal).

The reviewer is right, this was supposed to be part of future work. In fact, during our revisions made several failed attempts to triggering a delayed myosin contraction. One obvious idea might be to use photo-activatable caged ATP, however this comes with the complication that ATP is also needed for actin polymerization. In recent experiments we primarily tried to achieve myosin photo-activation through the myosin inhibitor blebbistatin. Blebbistatin is light sensitive, and it has been shown that the inhibitory effect can be turned off through exposure with blue light. However, only few studies have used this photo-dependent effect (*in vitro*) and unfortunately we were not able to reproduce these results.

Other comments –

- Are captions available for the movies? I don't seem to have them. The experimental conditions in each case should be given somewhere.

We now added captions to all movies in the supplemental PDF.

- For simulation of actin polymerization, it says in the SI that a fixed number of nuclei are created, and this is tuned to achieve a desired final length. I don't believe this initial nuclei number is listed in the paper for any of the simulation conditions. Does this number scale as a function of concentration in the cases where concentration is varied?

Yes, the number of nuclei is determined from the actin concentration and filament length. The number was tuned to give the same final filament at a given concentration and confinement size, as in Adeli Koudehi et al. We now include the number of nuclei in the figure captions.

- The authors should re-check the supplemental details on modeling for small typographical errors. For example, in the Langevin random force equation, the left hand side should have a t' , and in the attraction between actin filaments, the second r_i should be a vector(bold**), as well as those in the**

parentheses for the piecewise definition. Also, I believe the second order tensor $I_{\alpha\text{-beta}}$ refers to the independence of x,y, and z directions, (alpha and beta being axes), but this is not stated.

Thank you for catching these mistakes, which we have corrected, as well as clarifying that α and β label the x, y or z directions.

Reviewer #3 (Remarks to the Author):

Cell division facilitated by a contractile actomyosin ring is a crucial process for cell viability and understanding its biophysical details is critically important. Litschell et al. used synthetic biology to reconstruct ring-like actomyosin structures in giant unilamellar vesicles and provide a number of biophysical insights related to the formation, binding to the membrane and contraction of these structures. These results are confronted with a model to determine the key parameters that govern ring formation and membrane deformation.

The main new messages of this study are related to the importance of membrane binding on the probability of ring formation and the observation of membrane deformation following the constriction of the actin bundles. However, a number of concerns temper enthusiasm for the ongoing submission.

1/ I am uncomfortable with the way this manuscript is written, as it gives the impression that the scientific advances are more important than they actually are. In particular, it is astonishing that the landmark work of Miyazaki et al. (Nature Cell Biology, 2015) is not mentioned in the introduction, and only briefly referred to at the end of the discussion.

I believe that it is important to give credit to this milestone work for having reconstituted the formation of contractile actomyosin rings within liquid droplets, and for testing the significance of a large number of biophysical parameters that are important in this system. The authors should clarify unambiguously what the novelties of their study are. They should also highlight similarities and differences between both studies, particularly the relationship between compartment size and ring formation.

We apologize for giving the reviewer this negative impression. Our original thinking was to focus in the introduction on membrane based-systems and discuss the Miyazaki et al. paper in the discussion. However, we fully agree that this was not doing the importance of the work justice, and apologize for the misleading impression this could have given if not fixed. The work of Miyazaki et al. is one of the most important milestones towards reconstitution of contractile actomyosin rings *in vitro*, and as such very relevant for introducing our work.

We added the following section in the introduction:

“A true milestone towards the reconstitution of a division ring is the work of Miyazaki and coworkers.³³ They showed not only that the formation of equatorial rings from actin bundles is a spontaneous process that occurs in spherical confinement in order to minimize the elastic energy of the bundles, but also demonstrate the controlled constriction of these actomyosin rings.”

We added the following sections in our manuscript when presenting our results:

“Constriction of non-anchored actin rings was shown by Miyazaki et al., who demonstrated myosin-mediated contraction in a less cell-like system. They used actin bundled by depletion forces in water-in-oil droplets and showed that the behavior reproduced by this system has a striking resemblance to constricting cell division rings.³³”

We also still mention the work of Miyazaki et al in our discussion section of the paper and hope that the combination of all three sections is sufficient in the eyes of the reviewer.

2/ The length of actin filaments, which is a critical parameter to evaluate for the interpretation of these experiments, is only estimated based on a number of weak assumptions such as numbers of actin nuclei and rates of nucleation. It is not satisfying, because the manuscript does not indicate either the number of nuclei (which must be, I agree, quite hard to evaluate from pre-spun actin), or the numerical values obtained from this calculation.

I recommend the authors to use an independent way to measure the length of individual actin filaments. For example, stabilization, labeling and imaging with fluorescent phalloidin would provide a much better estimate of the length of individual actin filaments assembled from these protein mixtures. In addition, as length is a critical parameter in their model, it would be critical for this study to vary the length of actin filaments under controlled conditions, for example by addition of a protein such as capping protein which would induce the polymerization of much shorter filaments. Moreover, it is not always clear at first reading whether lengths 'L' correspond to the length of individual filaments or the length of the ring filament observed inside the vesicles. Could the author please clear up for each experiment what is the estimated length of individual actin filaments, the average length of cables (or ring diameter), the expected persistence length of the cables if crosslinkers are present and the size of vesicles measured?

The reviewer is correct in that our modeling work indicates that filament length is an important parameter when varied over distances comparable to the confining diameter. Investigating the effect of filament length using capping protein, profilin, etc. would indeed provide independent tests of the model. However, this would involve an extensive line of investigation that we feel is somewhat beyond our main result of reconstitution of contractile actomyosin rings in vesicles.

Further, we feel confident that the actin filament lengths under our conditions would be on the order of 6 μm , as demonstrated by many prior works, including the work by Miyazaki et al. We refer to this prior work now in the manuscript in the section about ring formation. Small changes of this value (filament length), in the range for example 3.5-10 μm , would not change the model results significantly. We hope that the reviewer is not confused by the simulation with short filaments in what is now Figure 4B; we used this result as an example of short filaments compared to the confining radius, and not as representing a possible filament length in our experiments (we did this because it is computationally expensive to perform simulations with both long filaments and even larger confining size).

Instead of trying to change filament length, a parameter that is easier to measure and naturally acquires different values is the size of the confining diameter. In this resubmitted version we show results for the probability of ring assembly as a function of vesicle size in Fig. S12. The agreement between theory and experiment is consistent with the expected value of individual filament length.

We also would like to note that, as expected, single filaments in our confocal microscopy assays are too thin to be resolved and distinguished from surrounding filaments and the background. In an unbundled state there are generally too many actin filaments to receive anything but an almost homogeneous signal (see fig. S3). In the bundled state filaments align with each other and thus blend even more into each

other. We attach below a negative stain electron microscopy image that we took as part of a different project, showing that, even with EM, filaments within a bundle cannot be distinguished in a way that would allow us to assess their length.

3/ It seems that there is a clear inhomogeneity in the amount of protein that is encapsulated in each vesicle. It is strikingly obvious in Figure S2 where neutravidin intensity varies widely between different vesicles of the same size. Could the authors provide a quantitative analysis of this variability, and tell how they took these variations into account when analyzing their results?

The reviewer is right, there is a significant variability of encapsulated protein in the vesicles. To quantify this, as the reviewer requested, we performed experiments in which we encapsulate a simple, soluble, fluorescent dye and analyzed the intensity distribution. We plotted a histogram for which we measured intensity of >1.000 GUVs, which we now include as Figure S4 in the supplements and also mention this in the main manuscript. The distribution of intensities follows a log normal distribution. To our knowledge, this result is typical for emulsion transfer methods. We assume that at some point during the vesicle formation process (most likely when the monolayered droplets pass through the final oil-water interphase) varying amounts of the surrounding aqueous solution get incorporated into the vesicles. Even though this seems inevitable and presumably affects all studies employing vesicle encapsulation, we are unaware of any literature discussing this in detail or quantifying the effect. We are aware of one publication in which a similar experiment is performed (Tsuji et al., PNAS, 2016, DOI: 10.1073/pnas.1516893113). In Figure 2C (black curve) of this publication a logarithmically plotted histogram of the intensity of a fluorescent marker within vesicles is shown (i.e. quite similar to our experiment which we just described). These vesicles are generated with the original emulsion transfer method and show an even broader intensity distribution. We include a logarithmically plotted histogram of our experiment in the same style as in the mentioned paper below, to allow for a better comparison.

We would like to point out that the vesicle generation method we use (cDICE) is a method that generates relatively homogeneous vesicles as compared to most other methods. This is especially the case when comparing to hydration methods, which suffer from variations in “encapsulation efficiency” and are the most common method for preparing GUVs.

4/ Vesicle size seems to be an important feature for the probability of ring formation, however the authors do not give any information about the vesicle size distribution analyzed in each experiment. Also it would be important to clarify the observation of Figure S2 and S3. The volume of vesicles being proportional to R^3 and their surface to R^2 , I would expect the percentage of membrane binding to be inversely proportional to both protein concentration and vesicle size. It doesn't seem to be the case. Could the authors clarify that?

We thank the reviewer for this comment, as it is a very important point and significantly affected one of the main findings of our work. Most of our data was generated from confocal images of single vesicles, which were restricted to specific size ranges. However, for Figure 2A (now Figure 4A) we analyzed large area scans with many vesicles. We have now plotted ring probability with respect to vesicle size, and restricted the data presented in Figure 2A (now Figure 4A) such that a specific size range is considered. We chose 15 to 20 μm , as a majority of vesicles considered for this study fall within that range. Finally, an analysis of ring formation probability with respect to size is now included in Figure S12.

The second part of the reviewer's comment refers to figure S2 and S3 and discusses the membrane coverage in regards to the volume to surface area ratio of the vesicles. We agree that the membrane coverage should, in theory, depend on the vesicle radius. (We are not sure what the reviewer means by “percentage of membrane binding”, but we would expect more membrane bound neutravidin with larger vesicles if sufficient biotinylated lipids are present.) However, as the reviewer also pointed out in point 3/, the concentrations vary between vesicles, as such this proportionality can sometimes not be directly

obvious.

5/ The deformation of the membrane vesicle by the actomyosin ring is a key outcome of this study, however the authors do not provide any quantification of this result nor information about its reproducibility. Could the authors include for example in the caption of figure 2D, the percentage of vesicles showing this effect in each experiment, vesicle size, membrane tension, and how many times the experiment was repeated?

We thank the reviewer for this comment, as it made us aware of some urgent changes of wording in our manuscript. We now put less emphasis on the membrane deformations and instead highlight the conclusions we can take from the slipping of the contractile membrane-anchored bundles. We also think that these conclusions reflect why it is difficult to achieve membrane deformations and particularly to catch the dynamic process with the microscope.

While we see many vesicles in the final contracted state, it is difficult to capture a dynamic contraction event, as we can only take time series of few vesicles per experimental run. Membrane deformation are a particularly unlikely event, and in fact, we think contracting structures in perfect ring-shapes immediately slip and do not deform the membrane at all, similar to the in vivo experiments of Stachowiak et al., which we mention in the manuscript.

We tried to highlight the problems that come with including myosin in our experiment system and hope our text makes a more transparent impression with phrases like the following:

“Although it appeared that the appropriate assay conditions for homogeneously contracting single rings have not yet been met in our giant vesicles, we recorded the constriction of a membrane-anchored ring-like structure along with membrane deformations.”

...and later in the same paragraph: *“Unfortunately the above mentioned complications in assay design hampered consistent observations of these membrane deformations.”*

As mentioned, in our discussion we added text explaining what conclusions can be drawn from our experiment to overcome the problems we observe in our experiment. We dedicate a new paragraph discussing what additional components need to be added and experimental parameters might need to be changed in order achieve a full vesicle division in future experiments.

In response to the reviewers comment, we also made some drastic changes to the papers structure and rearranged the order in which we present our results in the paper. We now present results regarding the effect of membrane binding of actin bundles before introducing actomyosin contractions. The actomyosin contractions which were previously shown in Figure 2, are now presented in the second to last figure, Figure 5. Along with some changes in wording, we think this now highlights our other results more. We also split the previous Figure 2 into two parts, which are now Figure 4 and Figure 5. We further include a second example (previously in the supplements) of actomyosin induced membrane deformation in the

current Figure 5 which hopefully gives the reader a better understanding of the diverse outcome of these experiments.

However, we think that although these experiments were difficult to reproduce, the unambiguous observation of few instances still represents a significant advance. We hope that the example shown in Figure 5A (previously Figure 2D) clearly and convincingly shows the reconstitution of all three functions: actin bundling, membrane anchoring and myosin contraction. While we experimentally struggled with the latter aspect, we think the example nicely demonstrates that the overall goal is a reachable one with membrane-bound actin bundles.

Minor comments

1/ Figure 2A presents the fraction of ring formation in the presence and absence of membrane binding; however, I am wondering why the data for VASP were not included in this analysis. Could the authors please include this information?

We did not perform experiments with VASP as extensively as with the other bundling proteins. However, we still thought it would demonstrate versatility to include it in Figure 1. [As also in in reply to point 4 by reviewer 2.]

There are also more practical reasons for the omission of VASP in Figure 2 (now Figure 4) due to the availability of the protein for us. The collaborator purifying this protein for us (Mizuno group), moved away.

We do think that showing three different bundling proteins is sufficient. If the reviewer perceives it as misleading and/or inconsistent to show VASP in Figure 1, but not in Figure 2 (now Figure 4), we would like to suggest to remove VASP from Figure 1 altogether.

2/ The authors mention the use of POPC lipids in the text and in Figure S1 while in the material and methods they indicate the use of DOPC

We apologize for this mistake. In this study we only used POPC (aside from small fractions of functionalized lipid).

3/ There are a couple of spelling and formatting issues. For example:

- **The authors write 'membraBe' instead of 'membrane' at multiple occasions**
- **In figure S8, the authors give reference to figure 2E instead of 2D**

We thank the reviewer for pointing out these typos in our figures.

4/ The authors do not mention figure s7 in the main text

We now do, as this figure was mentioned in the text we added in reply to major point 5/ by reviewer 3. Further, we moved this figure into the main text where it now is part of Figure 5.

5/ Could the author please include in the supporting material a more detailed description of how data were analyzed? This would include a description of the bundle curvature analysis shown in figure 3B, how the length of rings is analyzed in 3D, a description of how single clearly discernible ring were identified in vesicles, etc...

We apologize for submitting the manuscript in this condition, as it was lacking crucial information about methods and analysis. We now include a short section in the main text about our image analysis and more detailed descriptions in the supplements including how figure 3B (now Figure 2B) was made.

6/ Figure 2B: Could the authors present the first frame also without the yellow and orange lines?

The figure now shows the first frame without overlay, as requested. We also want to point out that all frames can be found without any overlays in the supplemental movie S6.

Reviewer #1 (Remarks to the Author):

Unfortunately, the marked changes document did not contain any marked changes (at least not visible to me, I apologize if this is my fault). Since the authors did not state the line numbers in the response, reviewing becomes very difficult.

Nevertheless, I appreciate the considerable amount of work that went into the revisions. The statistics are indeed impressive and the analysis has been thoroughly done. The account of existing literature is now fair. I (and certainly the scientific community) especially appreciate that the authors did not hesitate to share their code and the design of the cDICE chamber. With these additions, I think that the work is a significant advance and I am looking forward to see follow-up studies from different laboratories that validate and extend the results.

Reviewer #2 (Remarks to the Author):

In my previous comments on this work, I rated this work suitable for publication in Nature Communications, while adding a number of questions and suggestions that I felt would improve the manuscript. In the response, the authors have carefully taken the time to respond to each point. In particular they have clarified some aspects of the analysis that were confusing, and they have performed some of the simulations that I suggested.

Reading the comments of the other referees, it seems the main additional concerns were whether the authors gave enough credit to a prior important work, and how this manuscript was situated in the field of cytoskeletal reconstitution (as well as some similar questions to mine regarding how various analyses were performed). It seems the authors have reworked their manuscript to give due credit to prior studies, and to emphasize the advance being the reproducibility of their setup, as well as the ability to study membrane associated/bound actin filaments. In addition, they have addressed a number of other reviewer concerns, notably including one of their analysis scripts and CAD drawings for their setup, which will help enable future studies that build on this work.

Given that the authors have improved the manuscript, and addressed most comments (and it seems their extensive response may be published along with the article), I feel this work should be published in Nature Communications.

Glen Hocky
New York University

Reviewer #3 (Remarks to the Author):

I was skeptical about the originality of this work after reading the first version of the manuscript. The second version of the manuscript received marginal improvements to correct some analysis issues, statistical information and missing references. I have no doubt about the robustness of the experimental protocol, but it does not change the fact that this work is incremental compared to previous literature. Many comments from Reviewer 1 indicate that he/she had a pretty similar opinion of this work.

The problem is that most of the manuscript describes the formation of actin rings in cell-like

volumes, a reconstitution that has been done previously by other groups. The model is purely descriptive and does not provide any real new scientific knowledge. The only real novelty is in the last part of the manuscript (Figures 5 and 6) which shows, from still images, the deformations of the vesicles from myosin motor activity. Unfortunately, this part of the manuscript is not analyzed in depth, and does not add much to our understanding of these phenomena.

Overall, this manuscript remains far too preliminary in its current form, and should rather be presented as a method paper in a specialized journal.

Major comment:

In my opinion, there is still a significant problem in the analysis of these data. As I pointed out previously, protein encapsulation is indeed very inhomogeneous. The authors rightly explain that this phenomenon is inherent to the encapsulation process, but I still do not understand how this variability is taken into account in the current manuscript. Everything is written as if vesicles within a size range were analyzed regardless of their content, although variations in protein concentrations apparently vary by a factor of 1 to 15. How are the vesicles presented in the images selected? Are these variations in concentration leading to differences in the actin networks formed? Are numerical simulations correctly modeling these effects?

Minor comment:

Membrane is still not corrected in all Figures.

Reviewer #1 (Remarks to the Author):

Unfortunately, the marked changes document did not contain any marked changes (at least not visible to me, I apologize if this is my fault). Since the authors did not state the line numbers in the response, reviewing becomes very difficult.

Nevertheless, I appreciate the considerable amount of work that went into the revisions. The statistics are indeed impressive and the analysis has been thoroughly done. The account of existing literature is now fair. I (and certainly the scientific community) especially appreciate that the authors did not hesitate to share their code and the design of the cDICE chamber. With these additions, I think that the work is a significant advance and I am looking forward to see follow-up studies from different laboratories that validate and extend the results.

We apologize for the technical problems and for not referring to page numbers and lines in our reviewed document. We would also like to thank the reviewer for their valuable input and positive response to our final manuscript.

Reviewer #2 (Remarks to the Author):

In my previous comments on this work, I rated this work suitable for publication in Nature Communications, while adding a number of questions and suggestions that I felt would improve the manuscript. In the response, the authors have carefully taken the time to respond to each point. In particular they have clarified some aspects of the analysis that were confusing, and they have performed some of the simulations that I suggested.

Reading the comments of the other referees, it seems the main additional concerns were whether the authors gave enough credit to a prior important work, and how this manuscript was situated in the field of cytoskeletal reconstitution (as well as some similar questions to mine regarding how various analyses were performed). It seems the authors have reworked their manuscript to give due credit to prior studies, and to emphasize the advance being the reproducibility of their setup, as well as the ability to study membrane associated/bound actin filaments. In addition, they have addressed a number of other reviewer concerns, notably including one of their analysis scripts and CAD drawings for their setup, which will help enable future studies that build on this work.

Given that the authors have improved the manuscript, and addressed most comments (and it seems their extensive response may be published along with the article), I feel this work should be published in Nature Communications.

Glen Hocky
New York University

We thank the reviewer for the positive comments and appreciate the original feedback, which helped to improve the manuscript.

Reviewer #3 (Remarks to the Author):

I was skeptical about the originality of this work after reading the first version of the manuscript. The second version of the manuscript received marginal improvements to correct some analysis issues,

statistical information and missing references. I have no doubt about the robustness of the experimental protocol, but it does not change the fact that this work is incremental compared to previous literature. Many comments from Reviewer 1 indicate that he/she had a pretty similar opinion of this work.

The problem is that most of the manuscript describes the formation of actin rings in cell-like volumes, a reconstitution that has been done previously by other groups. The model is purely descriptive and does not provide any real new scientific knowledge. The only real novelty is in the last part of the manuscript (Figures 5 and 6) which shows, from still images, the deformations of the vesicles from myosin motor activity. Unfortunately, this part of the manuscript is not analyzed in depth, and does not add much to our understanding of these phenomena.

Overall, this manuscript remains far too preliminary in its current form, and should rather be presented as a method paper in a specialized journal.

Major comment:

In my opinion, there is still a significant problem in the analysis of these data. As I pointed out previously, protein encapsulation is indeed very inhomogeneous. The authors rightly explain that this phenomenon is inherent to the encapsulation process, but I still do not understand how this variability is taken into account in the current manuscript. Everything is written as if vesicles within a size range were analyzed regardless of their content, although variations in protein concentrations apparently vary by a factor of 1 to 15. How are the vesicles presented in the images selected? Are these variations in concentration leading to differences in the actin networks formed? Are numerical simulations correctly modeling these effects?

The main concern of the reviewer seems to be whether "vesicles within a size range were analyzed regardless of their content", to which we are happy to say that they were analyzed regardless of their content - with exception of clear outliers, which we specified more precisely in the supplementary information: "When characterizing actin bundle morphologies, we analyzed vesicles regardless of their content with exception of clear outliers, such as deformed vesicles and vesicles that did not contain any discernable actin bundles."

The concern of the reviewer seems to be reinforced by our experiment in the supplement, which we added as part of the revisions, in which we encapsulate a soluble dye. We can say however, that this effect does not seem to affect the outcome of our experiments and our analysis profoundly. We think it is important to include the figure, as the effect might need to be considered for some experiments and can be of interest to those wishing to use the approach for other systems. We can offer an explanation for why this seems to not affect our experiments noticeably, which we added to the Methods section: "We assume that this effect is reduced for a reaction mix containing actin in the process of polymerizing and bundling. During vesicle generation, this vesicle content is much less diffusive and therefore less likely to leave the vesicles."

The reviewer says they "have no doubt about the robustness of the experimental protocol", however, the main concern does seem to relate to the consistency of our experiments, asking whether there are "differences in the actin networks formed". We think our paper offers a high level of transparency in this regard. We offer a large palette of microscopy data with large fields of view that show tens to hundreds of vesicles. Therefore we not only show with our analysis the level of consistency of our data, but we also give insights into that by showing these random arrays of datapoints (vesicles).

Minor comment:

Membrane is still not corrected in all Figures.

We thank the reviewer for pointing this out once more. We hope we found the remaining instances of this typo.